# Transcriptional Repression of CCL2 by K_Ca_3.1 K^+^ Channel Activation and LRRC8A Anion Channel Inhibition in THP-1-Differentiated M_2_ Macrophages

**DOI:** 10.3390/ijms26157624

**Published:** 2025-08-06

**Authors:** Miki Matsui, Junko Kajikuri, Hiroaki Kito, Yohei Yamaguchi, Susumu Ohya

**Affiliations:** Department of Pharmacology, Graduate School of Medical Sciences, Nagoya City University, Nagoya 467-8601, Japan; c241739@ed.nagoya-cu.ac.jp (M.M.); kajikuri@med.nagoya-cu.ac.jp (J.K.); kito@med.nagoya-cu.ac.jp (H.K.); y_yamagu@med.nagoya-cu.ac.jp (Y.Y.)

**Keywords:** CCL2, K_Ca_3.1, LRRC8A, tumor-associated macrophage, tumor microenvironment, extracellular potassium ions (K^+^)

## Abstract

We investigated the role of the intermediate-conductance, Ca^2+^-activated K^+^ channel K_Ca_3.1 and volume-regulatory anion channel LRRC8A in regulating C-C motif chemokine ligand 2 (CCL2) expression in THP-1-differentiated M_2_ macrophages (M_2_-MACs), which serve as a useful model for studying tumor-associated macrophages (TAMs). CCL2 is a potent chemoattractant involved in the recruitment of immunosuppressive cells and its expression is regulated through intracellular signaling pathways such as ERK, JNK, and Nrf2 in various types of cells including macrophages. The transcriptional expression of CCL2 was suppressed in M_2_-MACs following treatment with a K_Ca_3.1 activator or an LRRC8A inhibitor via distinct signaling pathways: ERK–CREB2 and JNK–c-Jun pathways for K_Ca_3.1, and the NOX2–Nrf2–CEBPB pathway for LRRC8A. Under in vitro conditions mimicking the elevated extracellular K^+^ concentration ([K^+^]_e_) characteristic of the tumor microenvironment (TME), CCL2 expression was markedly upregulated, and this increase was reversed by treatment with them in M_2_-MACs. Additionally, the WNK1–AMPK pathway was, at least in part, involved in the high [K^+^]_e_-induced upregulation of CCL2. Collectively, modulating K_Ca_3.1 and LRRC8A activities offers a promising strategy to suppress CCL2 secretion in TAMs, potentially limiting the CCL2-induced infiltration of immunosuppressive cells (TAMs, T_reg_s, and MDSCs) in the TME.

## 1. Introduction

Tumor-associated macrophages (TAMs), which predominantly exhibit M_2_-like characteristics, are recruited to and infiltrate tumors in the tumor microenvironment (TME) via chemotactic factors derived from tumor and immune cells, including C-C motif Chemokine Ligand 2 (CCL2). M_2_-polarized TAMs influence tumor progression, metastasis, and immune evasion, and a high density of them is associated with poor patient survival in many cancer types [1,2]. Accordingly, strategies aimed at inhibiting M_2_-polarized TAM recruitment are currently under investigation. Understanding the molecular mechanisms that drive M_2_ polarization of TAM function in the TME is essential for the development of effective anti-cancer therapies.

CCL2, also known as monocyte chemoattractant protein 1 (MCP1), is produced not only by various types of cancer cells but also by immunosuppressive cells such as TAMs and myeloid-derived suppressor cells (MDSCs). CCL2 plays a pivotal role in cancer progression and metastasis [3,4]. Elevated levels of CCL2 are associated with poor overall survival in several cancer types [5,6], as it functions as a tumor-promoting factor. It is a key chemokine responsible for the recruitment and accumulation of TAMs, regulatory T cells (T_reg_s), and MDSCs within the TME [6,7]. Therefore, targeting CCL2 signaling represents a promising strategy for the development of tumor immunotherapies.

Potassium (K^+^) channels are involved in regulating cell proliferation, differentiation, migration, and metastasis [8]. In immune cells, the voltage-dependent K^+^ channel K_V_1.3 and intermediate-conductance, Ca^2+^-activated K^+^ channel K_Ca_3.1 are predominantly expressed. Their activation leads to K^+^ efflux, resulting in membrane hyperpolarization and increases Ca^2+^ influx through voltage-insensitive Ca^2+^ channels [8]. In the TME of solid tumors, necrosis elevates extracellular K^+^ concentration ([K^+^]_e_) [9], which in turn increases intracellular K^+^ levels and inhibits pro-inflammatory cytokine expression in cytotoxic T cells [9]. We have previously shown that exposure to high [K^+^]_e_ upregulates IL-10 and IL-8 expression in M_2_-polarized macrophages and that this effect is reversed by K_Ca_3.1 activation via the extracellular signal-regulated kinase (ERK) and c-Jun N-terminal kinase (JNK) signaling pathways [10].

Leucine-rich, repeat-containing protein 8A (LRRC8A) is a key component of the volume-regulated anion channel (VRAC), which is essential for regulating cell volume, osmotic balance, and intracellular signaling [11,12]. Recent studies have demonstrated that LRRC8A plays a crucial role in the activation of both T cells and macrophages [12,13]. We also previously reported that inhibition of LRRC8A suppresses IL-10 and IL-8 transcription in M_2_-polarized macrophages via the NADPH oxidase 2 (NOX2)–NF-E2-related factor 2 (Nrf2)–CCAAT enhancer-binding protein β (CEBPB) signaling axis [14].

The TME is characterized by altered ion concentrations, such as elevated [K^+^]_e_ [9]. Ionic shifts through ion channels and transporters can influence the behavior of M_2_-polarized TAMs by modulating intracellular signaling pathways. Ionic remodeling within the TME may contribute to CCL2, promoting the recruitment of monocytes and the maintenance of a pro-tumoral microenvironment. We previously demonstrated that K_Ca_3.1 activation and LRRC8A inhibition downregulated the immunosuppressive cytokine IL-10 in M_2_-MACs through the ERK–CREB, JNK–c-Jun, and NOX2–ROS–Nrf2–CEBPB signaling pathways, respectively [10,14]. CCL2 expression is also regulated through the ERK–CREB2, JNK, and Nrf2 transcriptional axes in various types of cells including macrophages [15,16,17,18]. Our primary aim of this study is to elucidate the potential involvement of K_Ca_3.1 and LRRC8A in regulating CCL2 expression and production through these signaling axes in M_2_-MACs, as well as their role in CCL2 upregulation under high-[K^+^]_e_ conditions that mimic the tumor microenvironment. Our findings suggest novel therapeutic strategies for blocking CCL2 signaling to inhibit the recruitment of immunosuppressive cells, including TAMs, into tumors.

## 2. Results

### 2.1. Decreased Expression and Secretion of CCL2 by K_Ca_3.1 Activation and LRRC8A Inhibition in M_2_-MACs

Tumor-associated macrophages (TAMs) are characterized as CD11b^+^CD163^+^ARG1^high^ cells. As previously reported, M_2_-MACs in this study exhibited high expression of M_2_ marker genes, including CD163, arginase 1 (ARG1), and C-C motif chemokine 22 (CCL22) [10]. CCL2 is a key chemokine involved in recruiting TAMs and other immunosuppressive cells, contributing to tumor growth, angiogenesis, and metastasis.

We investigated the transcriptional regulation of CCL2 in M_2_-MACs by K_Ca_3.1 activation and LRRC8A inhibition. As shown in Figure 1A,B, CCL22 was highly expressed (over 1000-fold of native THP-1 cells) and CCL2 was moderately expressed (approximately 5-fold) in M_2_-MACs. Treatment with the selective K_Ca_3.1 activator SKA121 (10 μM for 12 h) significantly reduced CCL2 transcript levels (Figure 1D), while having no significant effect on CCL22 expression (Figure 1C). Additionally, CCL2 secretion was significantly decreased following 24 h of SKA121 treatment (*p* < 0.01) (Figure 1E). Likewise, the treatment with the LRRC8A inhibitor endovion (EDV, 10 μM for 12 h) significantly reduced both CCL2 expression and secretion (Figure 1G,H), without affecting CCL22 expression (*p* < 0.01 and *p* > 0.05, respectively) (Figure 1F). Although EDV is also known to inhibit the Ca^2+^-activated Cl^−^ channel ANO1, our previous study confirmed that ANO1 is not functionally expressed in M_2_-MACs [14]. To further validate the role of LRRC8A, we conducted siRNA-mediated inhibition experiments. CCL2 expression but not CCL22 was significantly reduced by LRRC8A-specific siRNAs (*p* < 0.01) (Appendix A), with a knockdown efficiency of approximately 60% (Appendix A). These results indicate that both K_Ca_3.1 activation and LRRC8A inhibition suppress the expression and secretion of CCL2 in M_2_-MACs, thereby potentially limiting the recruitment of immunosuppressive cells such as TAMs and MDSCs into tumors within the TME.

### 2.2. Increased Expression and Secretion of CCL2 by High [K^+^]_e_ Exposure in M_2_-MACs and Suppressive Effects of K_Ca_3.1 Activation and LRRC8A Inhibition

In the hypoxic TME, the necrotic cancer cells release K^+^ ions, leading to elevated extracellular K^+^ concentration ([K^+^]_e_). An in vivo K^+^ imaging analysis showed that the K^+^ concentration was elevated in the TME, with an average concentration of approximately 29 mM [19]. In addition, the K^+^ concentration in tumor interstitial fluid was elevated up to 30–80 mM [9]. This increase in [K^+^]_e_ enhances immunosuppressive activity in the TME [8,20]. We previously reported that exposure to high [K^+^]_e_ (final K^+^ concentration: 35 mM) induces approximately twofold increases in IL-8 and IL-10 transcription in M_2_-MACs [10,14]. Interestingly, CCL2 transcript levels were increased more than 20-fold after 12 h of exposure to 35 mM [K^+^]_e_ in M_2_-MACs (*p* < 0.01) (Figure 2B). CCL2 secretion was also increased more than 10-fold after 24 h treatment (*p* < 0.01) (Figure 2C). The values of CCL2 concentration in control (−/−/−) and 35 mM [K^+^]_e_ (+/−/−) groups were 436.2 ± 2.5 and 5211.9 ± 29.2 pg/mL, respectively. The upregulation of CCL2 was comparable between 30 mM KCl (Figure 2B) and 15 mM K_2_HPO_4_ (Appendix A). In contrast, exposure to 5 mM MnCl_2_, 5 mM ZnSO_4_, or 30 mM NaCl had no significant effect on CCL2 mRNA levels (*p* > 0.05). Interestingly, exposure to 20 mM [Mg^2+^]_e_ by the addition of 19.6 mM MgSO_4_ significantly increased CCL2 transcript levels by approximately 10-fold (*p* < 0.01) (Appendix A). CCL2 was upregulated with increasing [K^+^]_e_ or [Mg^2+^]_e_ (Appendix A). Consistent with Figure 1D,G, the upregulating CCL2 induced by high [K^+^]_e_ was significantly suppressed by either SKA121 or EDV treatment for 12 h (*p* < 0.01) (Figure 2B,C). Furthermore, the combination treatment was more effective than either single treatment in suppressing CCL2 expression and secretion (*p* < 0.01).

In contrast, CCL22 transcript levels were unaffected by either single or combination treatments (*p* > 0.05) (Figure 2A). Intracellular K^+^ levels ([K^+^]_i_) were significantly increased (approximately 1.3-fold) by 35 mM [K^+^]_e_ exposure(*p* < 0.01) (Figure 2D), and this increase was largely reversed by SKA121 treatment (*p* < 0.01) (Figure 2D). In vehicle control, the intracellular K^+^ concentration ([K^+^]_i_) was 2.67 ± 0.05 mM/mg protein (*n* = 4), which is estimated to be between 140 and 150 mM. Similar effects of LRRC8A inhibition and high [K^+^]_e_ exposure on CCL2 expression were also observed in HL-60-differentiated M_2_-like macrophages (Appendix A).

### 2.3. Transcriptional Regulation via ERK, JNK, and Nrf2 Signaling Pathways in M_2_-MACs

We previously demonstrated that K_Ca_3.1 activation and LRRC8A inhibition downregulated the immunosuppressive cytokine IL-10 in M_2_-MACs through the ERK–CREB, JNK–c-Jun, and NOX2–ROS–Nrf2–CEBPB signaling pathways, respectively [10,14]. In addition, CCL2 expression is regulated through the ERK–CREB2, JNK, and Nrf2 transcriptional axes in various types of cells including macrophages [15,16,17,18]. Based on these findings, we examined whether these signaling axes are also involved in the CCL2 downregulation mediated by K_Ca_3.1 activation and LRRC8A inhibition. CCL2 expression levels were significantly reduced by treatment with the ERK inhibitor SCH772984 (1 μM, 12 h) (*p* < 0.01) (Figure 3A) and the JNK inhibitor SP600125 (1 μM, 12 h) (*p* < 0.01) (Figure 3B). The combination of these inhibitors was more effective than either single treatment in suppressing the high [K^+^]_e_-induced elevation of CCL2 expression and secretion (*p* < 0.01) (Figure 3C,D).

In addition, CCL2 expression was significantly reduced by treatment with the NOX2 inhibitor GSK1795039 (10 μM, 12 h) (*p* < 0.01) (Figure 3E) and the Nrf2 inhibitor ML385 (5 μM, 12 h) (*p* < 0.01) (Figure 3F). Either single treatment significantly suppressed the high [K^+^]_e_-induced increase in CCL2 expression (*p* < 0.01) (Figure 3G,H). Moreover, combination treatment with SCH772984, SP600125, and GSK1795039 was more effective than either SCH772984 plus SP600125 or GSK1795039 alone in suppressing high [K^+^]_e_-induced CCL2 expression and secretion (*p* < 0.01) (Figure 3I,J).

To investigate whether high [K^+^]_e_ exposure promotes nuclear translocation of phosphorylated Nrf2 (P-Nrf2), we visualized P-Nrf2 and unphosphorylated Nrf2 (Nrf2) in M_2_-MACs 2 h after exposure using laser scanning confocal fluorescence microscopy. P-Nrf2 and Nrf2 were stained with Alexa Fluor 488-conjugated secondary antibodies, and nuclei were counterstained with DAPI (Figure 4B,D). The relative fluorescence intensity of nuclear Alexa488-P-Nrf2 signals was significantly increased in M_2_-MACs exposed to high [K^+^]_e_ (*n* = 6, *p* < 0.01) (Figure 4C), whereas the intensity of cytosolic Alexa488-Nrf2 signals did not differ between vehicle- and high [K^+^]_e_-exposed cells (*n* = 6, *p* > 0.05) (Figure 4E). Control staining with Alexa Fluor 488-conjugated, anti-rabbit IgG secondary antibody alone showed minimal background fluorescence under the same imaging conditions (Figure 4A). These findings suggest that the transcriptional repression of CCL2 by K_Ca_3.1 activation and LRRC8A inhibition share a common mechanism with that of IL-10 in M_2_-MACs.

### 2.4. Possible Involvement of Transcriptional Regulators CEBPB and CREB2 in High [K^+^]_e_-Induced Upregulation of CCL2 in M_2_-MACs

In the present study, we found that exposure of M_2_-MACs to high [K^+^]_e_, mimicking the TME, resulted in more than a 20-fold increase in CCL2 transcriptional levels. However, the underlying mechanisms responsible for this upregulation remain unclear. Our previous studies showed that CEBPB and cyclic adenylic acid (cAMP) response element-binding protein (CREB) are involved in the transcriptional repression of IL-10 via LRRC8A inhibition and K_Ca_3.1 activation, respectively [10,14]. Therefore, we first investigated whether CEBPB contributes to high [K^+^]_e_-induced CCL2 upregulation in M_2_-MACs (Figure 5). Since no commercial CEBPB inhibitors are available, we employed CEBPB-specific siRNAs for gene knockdown. The knockdown efficiency of CEBPB siRNAs was approximately 55% in M_2_-MACs (Figure 5A), and siRNA-mediated CEBPB inhibition significantly suppressed high [K^+^]_e_-induced CCL2 expression (*p* < 0.01) (Figure 5B). However, neither transcript nor protein levels of CEBPB were significantly altered by high [K^+^]_e_ exposure (*p* > 0.05) (Figure 5C–E), suggesting that CEBPB may be functionally involved in CCL2 regulation without changes in its expression level. We next investigated the potential involvement of CREB family members in the high [K^+^]_e_-induced upregulation of CCL2 in M_2_-MACs (Figure 6). Under normal [K^+^]_e_ conditions, treatment with the pan-CREB inhibitor 666-15 (1 μM, 12 h) significantly reduced CCL2 transcript levels (*p* < 0.01) (Figure 6A). To identify the dominant CREB isoforms, we assessed the expression levels of nine isoforms: CREB1, CREB2 (ATF4), CREB3, CREB3L1, CREB3L2, CREB3L3, CREB3L4, CREB5, and ATF2. Among these, CREB2 was predominantly expressed in M_2_-MACs (Figure 6B), whereas CREB3L1, 3L3, 3L4, and 5 showed minimal expression (<0.001 arbitrary units). Importantly, both transcript and protein levels of CREB2 were significantly increased by high [K^+^]_e_ exposure (*n* = 4, *p* < 0.01) (Figure 6C–E). Furthermore, high [K^+^]_e_-induced increases in CCL2 expression and secretion were significantly suppressed by 666-15 (*p* < 0.01) (Figure 6F,G). These findings suggest that the ERK–CREB2 axis may contribute to the high [K^+^]_e_-induced upregulation of CCL2 in M_2_-MACs.

### 2.5. Possible Involvement of the WNK1–AMPK–p38 MAPK Signaling Pathway in High [K^+^]_e_-Induced Upregulation of CCL2 in M_2_-MACs

With-no-lysine (WNK) kinase 1 (WNK1) is a Cl^−^-sensing protein kinase known as an upstream negative regulator of AMP-activated protein kinase (AMPK) [21,22]. Among the four WNK isoforms (WNK1-4), WNK1 was predominantly expressed in M_2_-MACs, as reported previously [14]. The WNK1–AMPK–Nrf2 signaling axis has been identified as a key upstream pathway regulating cytokine expression [23,24]. However, we previously demonstrated that LRRC8A inhibition did not alter the phosphorylation level of AMPK in M_2_-MACs [14]. Consistently, LRRC8A inhibition also had no significant effect on the phosphorylation level of WNK1 in M_2_-MACs (*p* > 0.05) (Figure 7A,B), indicating that CCL2 suppression via LRRC8A inhibition is independent of the WNK1–AMPK–Nrf2 axis.

Interestingly, WNK1 activity is regulated not only by [Cl^−^]_i_ but also by [K^+^]_i_ [25]. Exposure to high [K^+^]_e_ significantly decreased the phosphorylation level of WNK1 (*p* < 0.01) (Figure 7C,D), while concurrently increasing the phosphorylation of AMPK (*p* < 0.01) (Figure 7E,F). Moreover, high [K^+^]_e_-induced CCL2 upregulation was significantly enhanced by treatment with the WNK1 inhibitor WNK-IN-11 (1 μM, 12 h) and the AMPK activator D942 (10 μM) (*p* < 0.01) (Figure 7G,I). In contrast, treatment with the AMPK inhibitor BAY3827 (1 μM) significantly suppressed CCL2 expression (*p* < 0.01) (Figure 7H). Since p38 MAPK is a known downstream target of AMPK, and Nrf2 can be activated through the AMPK–p38 MAPK–Nrf2 axis in M_2_-MACs [26], we also tested the effect of a p38 MAPK inhibitor. Treatments with PD169316 (1 μM, 12 h) significantly reduced high [K^+^]_e_-induced CCL2 upregulation (*p* < 0.01) (Figure 7J). Collectively, these results suggest that the WNK1–AMPK–p38 MAPK axis is involved in the transcriptional upregulation of CCL2 in response to high [K^+^]_e_ in M_2_-MACs.

### 2.6. Possible Involvement of Epigenetic Modification by HDAC3 in High [K^+^]_e_-Induced Upregulation of CCL2 in M_2_-MACs

Previous studies have reported that CCL2 expression is epigenetically regulated by histone deacetylases (HDACs) such as HDAC1, HDAC2, HDAC3, HDAC11, and SIRT1 in various types of immune cells [27,28]. In this study, we confirmed that HDAC1, 2, 3, and SIRT1 are expressed at relatively high levels in M_2_-MACs (Figure 8A), whereas HDAC11 expression was negligible (<0.001 arbitrary units). To evaluate their functional relevance, we tested specific inhibitors. High [K^+^]_e_-induced increases in CCL2 expression and secretion were significantly suppressed by treatment with the selective HDAC3 inhibitor T247 (10 μM) (*p* < 0.01) (Figure 8B,C). In contrast, treatment with the HDAC1/2 inhibitor AATB (10 μM) or the SIRT1 inhibitor Ex527 (10 μM) had no significant effect (*p* > 0.05). We further examined whether high [K^+^]_e_ affects HDAC3 expression. No significant change was observed in HDAC3 protein levels following high [K^+^]_e_ exposure (*p* > 0.05) (Figure 8D,E). In addition, HDAC3 transcript levels were not altered by K_Ca_3.1 activation or LRRC8A inhibition for 12 h (Figure 8F). These findings suggest that HDAC3 activity, rather than its expression level, may be involved in the epigenetic regulation of CCL2 under high [K^+^]_e_ conditions in M_2_-MACs.

### 2.7. Increased Expression and Secretion of CCL2 by High [Mg^2+^]_e_ Exposure in M_2_-MACs and Suppressive Effects of K_Ca_3.1 Activation and LRRC8A Inhibition

Magnesium ions (Mg^2+^) are essential for numerous physiological processes, including (1) cell proliferation and division, (2) energy metabolism, and (3) intracellular signal pathways [29]. In addition, Mg^2+^ has been shown to enhance immune responses against cancer [30,31]. In macrophages, Mg^2+^ promotes M_2_ polarization and upregulates M_2_ markers [32]. Notably, the Mg^2+^ concentration in RPMI 1640 medium is 10 mg/L (approximately 0.4 mM). As shown in Figure 9A,B, exposure to 19.6 mM MgSO_4_ (final extracellular Mg^2+^ concentration, [Mg^2+^]_e_: 20 mM) for 12 h resulted in more than a 10-fold increase in CCL2 transcript levels in M_2_-MACs (*p* < 0.01) (Figure 9A). Correspondingly, CCL2 secretion increased more than 5-fold after 24 h (*p* < 0.01) (Figure 9B). Similarly to the effects of high [K^+^]_e_ exposure, the high [Mg^2+^]_e_-induced upregulation of CCL2 was significantly suppressed by treatment with either SKA121 or EDV for 12 h (*p* < 0.01), and combination treatment was more effective than either alone (Figure 9A,B). We next examined candidate Mg^2+^ transporters in M_2_-MACs. Among eight candidates (TRPM6, TRPM7, MAGT1, SLC41A1, CNNM1-4), TRPM7 and MAGT1 transcripts were significantly upregulated during differentiation from native THP-1 to M_0_-MACs (Figure 9C,D). Notably, only MAGT1 was further upregulated during M_0_-to-M_2_ polarization (Figure 9D). Other candidates were minimally expressed (<0.01 arbitrary units). Since no commercial MAGT1 inhibitors are available, we employed MAGT1-specific siRNAs for gene knockdown. The knockdown efficiency of MAGT1 siRNAs was approximately 40% in M_2_-MACs (Figure 9E), and siRNA-mediated MAGT1 inhibition significantly suppressed high [Mg^2+^]_e_-induced CCL2 upregulation (*p* < 0.01) (Figure 9F). These results suggest that MAGT1 may play a crucial role in Mg^2+^ transport in M_2_-MACs, thereby regulating CCL2 transcription.

Our previous studies showed that high [K^+^]_e_ enhances IL-8 and IL-10 expression and secretion via activation of ERK, JNK, and Nrf2 pathways in M_2_-MACs [10,14]. Similarly, recent reports indicate that Mg^2+^ promotes IL-10 secretion and induces JNK phosphorylation in THP-1-differentiated macrophages [32,33]. Consistent with these findings, exposure to high [Mg^2+^]_e_ (20 mM) approximately doubled the expression of IL-8 and IL-10 in M_2_-MACs. This upregulation was significantly suppressed by LRRC8A inhibition with 10 μM EDV and K_Ca_3.1 activation with 10 μM SKA121 (Figure 9G–J). Similarly, IL-8 and IL-10 secretion levels were also significantly decreased by either treatment (*p* < 0.01), with combination treatment being more effective in suppressing CCL2 secretion (*p* < 0.01) (Figure 9K,L). We further assessed the involvement of ERK, JNK, and Nrf2 pathways in high [Mg^2+^]_e_-induced CCL2 upregulation. Western blot analysis revealed that exposure to high [Mg^2+^]_e_ significantly increased the phosphorylation of ERK, JNK, and c-Jun in M_2_-MACs (*p* < 0.01) (Figure 10A–F). In addition, consistent with the high-[K^+^]_e_ experiments (Figure 4), the nuclear localization of P-Nrf2 was significantly increased following high [Mg^2+^]_e_ exposure (*n* = 6, *p* < 0.01) (Figure 10G,H), whereas cytosolic Nrf2 levels remained unchanged (*n* = 6, *p* > 0.05) (Figure 10I,J). These results suggest that like high [K^+^]_e_, high [Mg^2+^]_e_ upregulates CCL2 expression in M_2_-MACs via activation of the ERK, JNK, and Nrf2 signaling pathways, and this effect can be counteracted by K_Ca_3.1 activation and LRRC8A inhibition.

## 3. Discussion

Elevated levels of CCL2 correlate with increased TAM accumulation in the TME, enabling tumor cells to evade immune surveillance. High CCL2 expression is also associated with aggressive malignancies, increased metastatic potential, and poor prognosis in various cancers [5,6,7]. Strategies that block CCL2 signaling (i.e., the CCL2-CCR2 axis) inhibit TAM recruitment and reprogram their functions to support anti-tumor immunity. In the present study, we investigated the role of the K_Ca_3.1 K^+^ channel and LRRC8A Cl^−^ channel in regulating CCL2 expression in M_2_-MACs, which serve as a model for TAMs. Our findings highlight the critical roles of K_Ca_3.1 and LRRC8A in modulating CCL2 levels in M_2_-MACs under both normal and TME-mimicking ionic conditions (Figure 11). The key findings obtained in the present study are as follows: (1) Both K_Ca_3.1 activation and LRRC8A inhibition significantly downregulated CCL2 expression in M_2_-MACs, without affecting CCL22 expression (Figure 1). In addition to CCL2, CCL22, and IL-10, TAM-associated markers CCL5 and SPP1 (secreted phosphosprotein 1, also known as osteopontin, OPN) are promising targets for novel cancer immunotherapy [34,35]. Elevated expression of SPP1 and CCL5 in TAMs has been linked to worse clinical outcomes [34,36]. Although differentiation into M_2_-MACs increased their expression, neither K_Ca_3.1 activation nor LRRC8A inhibition affected their levels (Appendix A). (2) Mechanistically, K_Ca_3.1 activation suppressed CCL2 expression via the ERK–CREB2 and JNK–c-Jun pathways, whereas LRRC8A inhibition acted through the NOX2–Nrf2–CEBPB axis (Figure 3, Figure 4 and Figure 5). Notably, these pathways overlap with those involved in IL-10 downregulation, suggesting a shared transcriptional regulatory mechanism among multiple immunosuppressive mediators. The LRRC8A inhibition-mediated downregulation of CCL2 was an independent action from the WNK1-AMPK-Nrf2 axis (Figure 7). (3) Under elevated [K^+^]_e_ conditions mimicking the TME, we observed a dramatic upregulation of CCL2 levels in M_2_-MACs (Figure 2). This was accompanied by increased [K^+^]_i_, which was partially reversed by K_Ca_3.1 activation (Figure 2). Importantly, both K_Ca_3.1 activation and LRRC8A inhibition significantly attenuated high [K^+^]_e_-induced CCL2 upregulation (Figure 2, Figure 3, Figure 4 and Figure 5), further supporting their regulatory roles under tumor-like ionic conditions. We confirmed that all reagents and siRNAs used in this study did not cause any significant changes in cell viability (Appendix A).

CCL2 is subject to epigenetic regulation involving both DNA methylation and histone deacetylation [37]. We found that high [K^+^]_e_-induced CCL2 upregulation was significantly suppressed by HDAC3 inhibition (Figure 8) without changes in HDAC3 expression level upon high [K^+^]_e_ exposure, implicating HDAC3-mediated chromatin remodeling as a key regulator under high [K^+^]_e_ conditions and as a potential contributor to TAM recruitment in the TME. However, the effect of pharmacological inhibition of HDACs is indirect evidence. Further studies of direct measurement of histone deacetylation status (i.e., acetylated histone levels) and HDAC enzymatic activity will be needed to strengthen the mechanistic interpretation. Additionally, HDAC3 activity is known to be regulated through interactions with corepressors such as N-CoR (nuclear receptor corepressor) and SMRT (silencing mediator for retinoic acid and thyroid hormone receptors) [38,39], which have been implicated in HDAC3-dependent M_2_ polarization [40]. Further studies are required to elucidate the role of these corepressors in HDAC3-mediated CCL2 regulation in TAMs.

Previous reports have shown that nucleotide-binding oligomerization domain, leucine-rich repeat (NLR) family pyrin domain-containing 3 (NLRP3) is highly expressed in TAMs, and that nigericin-induced K^+^ efflux activates the NLRP3 inflammasome [41]. NLRP3 activation is also known to inhibit the ERK pathway [42]. Consistent with this, our study demonstrated that treatment with two NLRP3-activating K^+^ ionophores—nigericin and valinomycin (each at 10 μM)—significantly downregulated CCL2 expression under both normal and high [K^+^]_e_ conditions (*n* = 4, *p* < 0.01) (Appendix A). These findings suggest that NLRP3 may mediate both K_Ca_3.1 activation-induced downregulation and high [K^+^]_e_-induced enhancement of CCL2 expression. Additionally, since Cl^−^ efflux-induced reductions in [Cl^−^]_i_ promote NLRP3 activation [21], and activation of Cl^−^-sensing WNK1 signaling acts to suppress NLRP3 activation by balancing [Cl^−^]_i_ and [K^+^]_i_ [43], LRRC8A inhibition, which increases [Cl^−^]_i_, may inactivate NLRP3. However, we observed no change in P-WNK1 levels upon LRRC8A inhibition (Figure 7A,B), suggesting that this mechanism may operate independently of LRRC8A. Conversely, high [K^+^]_e_ exposure reduced WNK1 phosphorylation and enhanced AMPK phosphorylation and AMPK/p38 MAPK activation (Figure 7C–J), consistent with previous studies showing high K^+^ medium suppresses WNK1 activity and activates NLRP3 via AMPK phosphorylation [44,45]. Therefore, the NLRP3–WNK1–AMPK axis may be a novel pathway contributing to high [K^+^]_e_-induced CCL2 upregulation in M_2_-MACs.

Chemokines are post-transcriptionally regulated by short and non-coding microRNAs (miRNAs) in macrophages including TAMs [46]. Several miRNAs influence macrophage polarization [47]. For example, miR-511 downregulates CCL2 in M_2_-polarized TAMs [48]. Other studies have identified miR-192 as a TAM-enriched miRNA [49] and its expression was shown to reduce CCL2 levels in aged mice [50]. Notably, K^+^ ions reduce miR-192 expression in the kidney [51]. These observations suggest that these miRNAs may participate in both high [K^+^]_e_-induced CCL2 upregulation and its suppression by K_Ca_3.1 activation or LRRC8A inhibition. To validate this hypothesis, further investigations of their roles in M_2_-MACs and TAMs are required.

Recent work by Guo et al. (2024) demonstrated that Runt-related transcription factor 1 (RUNX1) promotes the recruitment of M_2_-polarized TAMs into the TME by enhancing CCL2 transcription [52]. RUNX1 functions in cooperation with transcriptional and epigenetic regulators such as HDAC3, CREB2, and CEBPB [53,54,55]. Intriguingly, RUNX1 expression was largely upregulated under high [K^+^]_e_ conditions (Appendix A). siRNA-mediated inhibition of RUNX1 significantly decreased the CCL2 mRNA level and secretion (Appendix A); however, no significant changes in HDAC3, CREB2, and CEBPB levels were observed (Appendix A). In addition, no significant changes in RUNX1 levels were observed upon K_Ca_3.1 activation and LRRC8A inhibition (Appendix A). To validate the possible involvement of RUNX1 in upregulation of CCL2 under high [K^+^]_e_ conditions—independent of HDAC3, CREB2, and CEBPB—in TAMs, further investigations are required.

Recent studies have highlighted the essential role of Mg^2+^ in enhancing the cytotoxicity of intratumoral CD8^+^ T cells and NK cells [56,57]. Consistent with this, patients with low serum Mg^2+^ levels exhibited reduced survival following CAR-T therapy [30]. However, the role of Mg^2+^ in the TME, particularly in immunosuppressive cells, remains largely unexplored. In the present study, we observed that elevated [Mg^2+^]_e_ similarly upregulated CCL2 expression (Figure 9). The effect was likely mediated by Mg^2+^ possibly via MAGT1 (Figure 9D). Mimicking the effects of high [K^+^]_e_, high [Mg^2+^]_e_ exposure activated ERK, JNK, and Nrf2 pathways (Figure 10), and high [Mg^2+^]_e_-induced CCL2 upregulation was reversed by K_Ca_3.1 activation and LRRC8A inhibition (Figure 9A,B). Additionally, high [Mg^2+^]_e_ exposure increased the expression levels of CREB2 but not CEBPB and HDAC3 (Appendix A), suggesting that elevated K^+^ and Mg^2+^ in the TME may converge on common signaling mechanisms that reinforce the immunosuppressive TAM phenotype. Cytotoxic T cells and TAMs coexist and interact dynamically within the TME. Therefore, further investigations using co-culture systems are essential to elucidate the role of Mg^2+^ in cancer immunotherapy.

Although IL-34 is known to reprogram macrophages into TAMs via the IL-34–CSF1R (colony-stimulating factor 1 receptor, known as IL-34 receptor) axis [58,59], we found that CSF1R expression in M_2_-MACs was extremely low (<0.01 arbitrary units, *n* = 4), and that IL-4/IL-13- and IL-4/IL-34-treated macrophages showed no significant differences in M_2_ marker expression (Appendix A). Despite the comprehensive scope of our analyses, several limitations remain. First, our study relied primarily on THP-1-derived M_2_ macrophages, which do not fully replicate the heterogeneity of in vivo TAMs. In vitro–differentiated M_2_ macrophages do not fully recapitulate the diverse intracellular signaling of TAMs and lack exposure to actual TME conditions such as hypoxia, elevated lactate, and low glucose. Second, although we identified key signaling pathways—including ERK–CREB2, JNK–c-Jun, NOX2–Nrf2–CEBPB, and WNK1–AMPK–p38 MAPK and an epigenetic regulation via HDAC3, their interconnectivity and temporal dynamics remain to be elucidated. In the present study, the knockdown efficiency of the siRNAs was relatively low. Future studies could employ more robust gene-editing approaches, such as CRISPR/Cas9-mediated knockout, to achieve more complete and stable gene disruption. This would enable a more comprehensive understanding of the target molecule’s function under the experimental conditions.

Collectively, our results suggest that modulating K_Ca_3.1 and LRRC8A activity offers a promising strategy to suppress CCL2 secretion in TAMs, potentially limiting the CCL2-induced infiltration of immunosuppressive cells (TAMs, T_reg_s, and MDSCs) into tumors. Targeting K_Ca_3.1 and LRRC8A, along with pathways such as WNK1–AMPK–p38 MAPK and HDAC3, may provide new therapeutic avenues to reeducate and reprogram TAMs within the TME, thereby enhancing the efficacy of cancer immunotherapy. In cancer types that functionally express K_Ca_3.1, activation of K_Ca_3.1 may potentially promote cellular proliferation and invasiveness. However, in solid tumors, non-malignant cells are often reported to comprise approximately 30–70% of the total tumor mass. While the proportion of non-cancerous cells varies greatly depending on the cancer type, stage, and so on, immunosuppressive cells, including TAMs, typically account for about 10–30%. Therefore, even in K_Ca_3.1-positive solid tumors, if immunosuppressive cells such as T_reg_s, TAMs, or MDSCs represent a significant portion of the tumor, activating K_Ca_3.1 may suppress the function of these immunosuppressive cells, restore immune surveillance, and thereby inhibit tumor growth. Additionally, several studies, which report findings that contrast with those of the present study, have demonstrated that K_Ca_3.1 inhibitors can attenuate tumor progression and improve clinical outcomes by modulating the phenotypic and functional properties of immune cells within the TME. In murine models of glioma, pharmacological inhibition of K_Ca_3.1 has been shown to reduce tumor burden by promoting a phenotypic shift in TAMs from a pro-tumorigenic M_2_ to an anti-tumorigenic M_1_ phenotype [60,61]. Similarly, in renal clear cell carcinoma, transcriptomic analyses of tumor-infiltrating immune cells have revealed a negative correlation between K_Ca_3.1 expression levels in tumor burden and patient prognosis [62]. These findings underscore the need for further investigation to delineate the specific cancer types and stages in which K_Ca_3.1 activators or LRRC8A inhibitors may exert therapeutic efficacy.

## 4. Materials and Methods

### 4.1. Materials and Reagents

The human acute monocytic leukemia cell line, THP-1, was supplied by Cell Resource Center for Biomedical Research, Tohoku University (Sendai, Japan). RPMI-1640 medium (189-02025) was purchased from FUJIFILM Wako Pure Chemicals (Osaka, Japan). Fetal bovine serum (Product code: 172012), phorbol 12-myristate 13-acetate (PMA) (P1585), DAPI (D2542), and Phosphatase Inhibitor Cocktails 2 (P5726)/3 (P0044) were from Sigma-Aldrich (St. Louis, MO, USA). Recombinant human IL-4 (AF-200-04), IL-13 (AF-200-13), and IL-34 (AF200-34) were from PeproTech (Cranbury, NJ, USA). Luna Universal qPCR Master Mix (M3003E) was from New England Biolabs Japan (Tokyo, Japan). EDV (HY-105917), SKA121 (HY-107414), SP600125 (HY-12041), and PD169316 (HY-10578) were from MedChemExpress (Monmouth Junction, NJ, USA). Human MCP-1/CCL2 Uncoated enzyme-linked immunosorbent assay (ELISA) kits (88-7399-88), Human IL-10/IL-8 Uncoated ELISA kits (88-7106-77, 88-8086-77), pre-designed siRNAs for CEBPB (siRNA ID: s2893), LRRC8A (s32108), MAGT1 (s224928), RUNX1 (s229351), Silencer Select Negative Control No. 1 siRNA (4390843), and SuperSignal West Pico PLUS Chemiluminescent Substrate (34580) were from Thermo Fisher Scientific (Waltham, MA USA). WNK-IN-11 (29676), SCH772984 (19166), D942 (14741), and GSK2795039 (33777) were from Cayman Chemical (Ann Arbor, MI, USA). ML385 (S8790), BAY3827 (S9833), 666-15 (S8846), and Ex527 (S1541) were from Selleckchem (Yokohama, Japan). A CytoFix/Perm kit was from BD Pharmingen (AB_2869008) (Franklin Lakes, NJ, USA). Potassium (K) Turbidimetric Assay Kit (E-BC-K279-M) was from Elabscience (Houston, TX, USA). PCR primers were from Nihon Gene Research Laboratories (Sendai, Japan). The HDAC inhibitors (vorinostat, AATB, and T247) were supplied by Professor Suzuki (Osaka University, Osaka, Japan). Other chemicals and reagents were from Sigma-Aldrich, FUJIFILM Wako Pure Chemicals, and Nacalai Tesque (Kyoto, Japan).

### 4.2. Cell Culture and Differentiation into M_2_-MACs

The differentiation of THP-1 and HL-60 into M_0_ MACs was induced by treatment with PMA (100 ng/mL) for 8 h. After removal of the medium, cells were incubated with RPMI 1640 medium supplemented with IL-4 and IL-13 (20 ng/mL for each) for 72 h to induce the polarization of M_2_ macrophages [10,14]. We previously confirmed that M_2_ markers such as CD163, arginase 1, and IL-10 were highly expressed 24 h after IL-4 and IL-13 supplementation and remained stably elevated up to 72 h [10,14]. Therefore, to align the endpoints for drug applications, we performed them at 60 h (for 12 h in real-time PCR assay) and at 48 h (for 24 h in Western blot and ELISAs).

### 4.3. Real-Time PCR

Total RNA extraction and cDNA synthesis were conducted as previously reported [14]. The gene-specific primers for real-time PCR examinations were designed using Primer Express^TM^ software (Version 1.5, Thermo Fisher Scientific). Real-time PCR was performed using Luna Universal qPCR Master Mix (New England Biolabs Japan, Tokyo, Japan) on the ABI 7500 Fast real-time PCR instrument (Applied Biosystems, Waltham, MA, USA) and analyzed with 7500 Fast System SDS software Version 1.5.1 [14]. PCR primers of human origin are listed in Appendix A. Unknown quantities relative to the standard curve for a particular set of primers were calculated [14], yielding the transcriptional quantitation of gene products relative to ACTB.

### 4.4. Western Blots

Whole-cell lysates were extracted by RIPA buffer. Phosphatase inhibitor cocktails were added to extracts at a final concentration of 1%. Equal amounts of protein were subjected to SDS-PAGE and immunoblotting, with the antibodies shown in Appendix A, and were then incubated with an anti-rabbit or anti-mouse HRP-conjugated IgG secondary antibody. An ECL advance chemiluminescence reagent kit was used to identify the bound antibody. The resulting images were analyzed using Amersham Imager 600 (GE Healthcare Japan, Tokyo, Japan). The optical density of the protein band signal relative to that of the ACTB signal was calculated using ImageJ software (Version 1.42, National Institutes of Health, Bethesda, MD, USA), and protein expression levels in the vehicle control were then expressed as 1.0. To assess the significance of differences between two groups, the paired Student’s *t*-test with Welch’s correction was used.

### 4.5. Measurement of Cytokine Production by ELISA

Human CCL2, IL-10, and IL-8 levels in supernatant samples were measured with the respective CCL2/IL-10/IL-8 Human Uncoated ELISA kits (Thermo Fisher Scientific), according to the manufacturer’s protocols. Standard curves were plotted using a series of cytokine/chemokine concentrations. Absorbance was measured using the microplate reader SpectraMax 384 (Molecular Devices Japan, Tokyo, Japan) at a wavelength of 450 nm (reference wavelength: 650 nm) with Soft Max Pro Software Version 5.0.

### 4.6. Measurement of [K^+^]_i_ by Potassium Turbidimetric Assay

Intracellular K^+^ contents were measured with Potassium (K) Turbidimetric Assay kit, according to the manufacturer’s protocols. The turbidity is proportional to the K^+^ concentration. Absorbance was measured using the microplate reader SpectraMax 384 (Molecular Devices Japan, Tokyo, Japan) at a wavelength of 450 nm (reference wavelength: 650 nm) with Soft Max Pro Software Version 5.0.

### 4.7. Confocal Imaging of the Nuclear and Cytosolic Distribution of Nrf2

After the treatment with trypsin solution, isolated M_2_-MACs were fixed and permeabilized using the CytoFix/Perm kit. The antibodies shown in Appendix A were labeled with an Alexa Fluor 488-conjugated secondary antibody. After seeding onto the glass bottom dish, fluorescence images were visualized using a confocal laser scanning microscope system (Nikon A1R, Tokyo, Japan) [14]. Image data were quantitatively analyzed using ImageJ software. For each experimental replicate (*n* = 1), fluorescence intensities were obtained from at least four frame images containing more than 30 cells. Summarized data were derived from six independently differentiated M_2_-MAC samples. Cell immunostaining, imaging acquisition, and quantitative analyses were independently performed by separate investigators to minimize potential bias.

### 4.8. Statistical Analysis

Statistical analyses were performed with the statistical software XLSTAT (version 2013.1). In Western blots (Section 4.4), the paired Student’s *t*-test with Welch’s correction was used. In the other experiments, to assess the significance of differences between two groups and among multiple groups, the unpaired Student’s *t*-test with Welch’s correction or one-way ANOVA with Tukey’s test was used. All experiments were performed with at least four independent biological replicates unless otherwise stated. Results with a *p*-value < 0.05 were considered to be significant. Data are shown as means ± standard error.

## 5. Conclusions

The present study uncovered a novel immunomodulatory role for ionic balances of [K^+^]_i_ and [Cl^−^]_i_ in regulating CCL2 expression in M_2_-MACs, which serve as a model for TAMs. To our knowledge, this is the first report demonstrating that elevated [K^+^]_e_ (also [Mg^2+^]_e_) robustly induces CCL2 transcription and secretion in M_2_-MACs, and that this induction can be reversed by K_Ca_3.1 activation and LRRC8A inhibition. These findings emphasize the fundamental role of ionic remodeling within the TME in TAM education. While the clinical utility of K_Ca_3.1 activators and LRRC8A inhibitors remains to be fully clarified, the present findings offer new insights that support their potential application in TAM-targeted cancer immunotherapies. Given the major role of CCL2 in immunosuppressive processes, combinatorial approaches involving ion channel modulators and CCR2 antagonists or immune checkpoint inhibitors may yield synergistic benefits in cancer treatment. CCL2 is a key component of the senescence-associated secretory phenotype (SASP), and CCL2-driven signaling is implicated in the development of age-related diseases. Further validation of the mechanisms by which K_Ca_3.1 and LRRC8A regulate CCL2 expression to reprogram TAMs may offer a promising and underexplored strategy for treating age-related diseases. The current findings, derived from in vitro experiments using THP-1-differentiated M_2_ macrophages, provide valuable insights into the regulatory mechanisms involved in CCL2 expression under tumor-mimicking ionic conditions. For the next stage, it will be important to assess whether similar regulatory pathways operate in primary human TAMs, co-culture systems involving tumor cells and macrophages, and in vivo tumor models that recapitulate the complexity of the TME. These validation studies will help to confirm the translational relevance of our findings and support their potential application in therapeutic strategies targeting TAM-mediated immunosuppression.

## Figures and Tables

**Figure 1 ijms-26-07624-f001:**
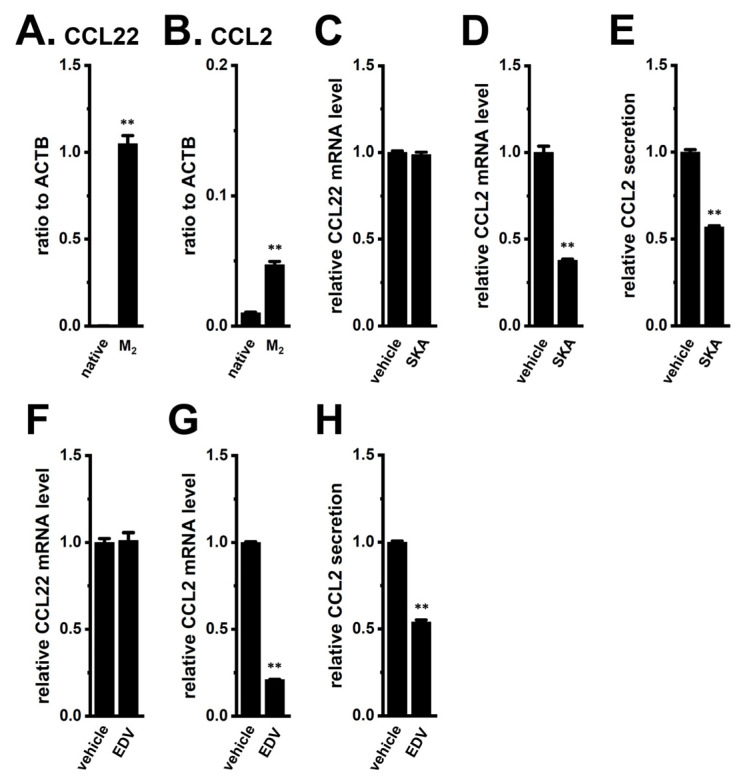
Effects of K_Ca_3.1 activation and LRRC8A inhibition on CCL2 expression and secretion in M_2_-MACs. (**A**,**B**): Real-time PCR examination of CCL22 (**A**) and CCL2 (**B**) expression in native THP-1 cells (native) and M_2_-MACs (M_2_) (*n* = 4). Data are normalized to ACTB. (**C**,**D**,**F**,**G**): Real-time PCR examination of CCL22 (**C**,**F**) and CCL2 (**D**,**G**) expression following treatment with vehicle, 10 μM SKA121 (SKA) (**C**,**D**), or 10 μM endovion (EDV) (**F**,**G**) for 12 h in M_2_-MACs (*n* = 4). Expression levels are normalized to ACTB and shown relative to vehicle control (set as 1.0). (**E**,**H**): ELISA quantification of CCL2 secretion after 24 h treatment with SKA (**E**) or EDV (**H**) in M_2_-MACs (*n* = 4). Secretion levels are shown relative to vehicle control (set as 1.0). **: *p* < 0.01 vs. native THP-1 or vehicle control.

**Figure 2 ijms-26-07624-f002:**
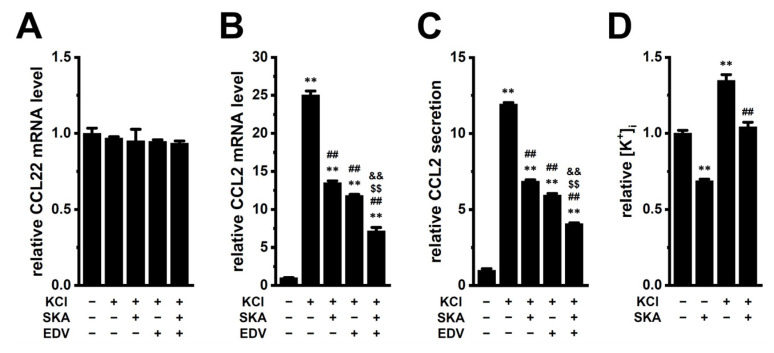
Upregulation of CCL2 by high [K^+^]_e_, as well as the suppressive effects of K_Ca_3.1 activation and LRRC8A inhibition on CCL2 expression and secretion in M_2_-MACs. (**A**,**B**): Real-time PCR examination of CCL22 (**A**) and CCL2 (**B**) expression in M_2_-MACs treated with vehicle (−/−/−), 30 mM KCl (+/−/−), and KCl + 10 μM SKA (+/+/−), +10 μM EDV (+/−/+), or + both (+/+/+) for 12 h (*n* = 4). The plus (+) and minus signs (−) are used to represent the presence and absence of the reagents, respectively. (**C**): ELISA quantification of CCL2 secretion from the same treatment groups as above for 24 h (*n* = 4). (**D**): Intracellular K^+^ concentration ([K^+^]_i_) measured by turbidimetric assay after 2 h treatment with vehicle (−/−), SKA (−/+), KCl (+/−), or KCl + SKA (+/+) (*n* = 4). Data are normalized to vehicle control. **: *p* < 0.01 vs. vehicle control (−/−/− or −/−); ##: *p* < 0.01 vs. +/−/− or +/−; $$: *p* < 0.01 vs. +/+/−; &&: *p* < 0.01 vs. +/−/+.

**Figure 3 ijms-26-07624-f003:**
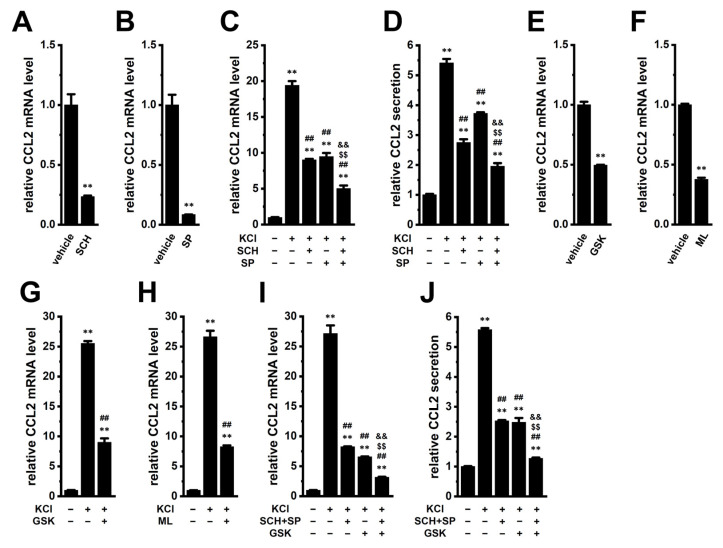
Inhibition of CCL2 expression by pharmacological blockade of ERK, JNK, NOX2, and Nrf2 in M_2_-MACs. (**A**,**B**): Real-time PCR examination of CCL2 expression in M_2_-MACs treated with vehicle, 1 μM SCH772984 (SCH; ERK inhibitor) (**A**), and 1 μM SP600125 (SP; JNK inhibitor) (**B**) for 12 h (*n* = 4). (**C**,**D**): CCL2 expression (**C**) and secretion (**D**) in M_2_-MACs treated with vehicle (−/−/−), 30 mM KCl (+/−/−), and KCl + SCH (+/+/−), + SP (+/−/+), or + both (+/+/+) (*n* = 4). (**E**,**F**): CCL2 expression following 12 h treatment with 10 μM GSK2795039 (GSK; NOX2 inhibitor) (**E**), and 5 μM ML385 (ML; Nrf2 inhibitor) (**F**) (*n* = 4). (**G**,**H**): CCL2 expression after 12 h treatment with KCl alone (+/−) or combined with GSK (**G**) or ML (**H**) (+/+) (*n* = 4). (**I**,**J**): CCL2 expression (**I**) and secretion (**J**) in M_2_-MACs treated with vehicle (−/−/−), KCl (+/−/−), KCl + SCH + SP (+/+/−), KCl + GSK (+/−/+), or KCl + all three inhibitors (+/+/+) (*n* = 4). **: *p* < 0.01 vs. vehicle control (−/−/− and −/−); ##: *p* < 0.01 vs. +/−/− or +/−; $$: *p* < 0.01 vs. +/+/−; &&: *p* < 0.01 vs. +/−/+.

**Figure 4 ijms-26-07624-f004:**
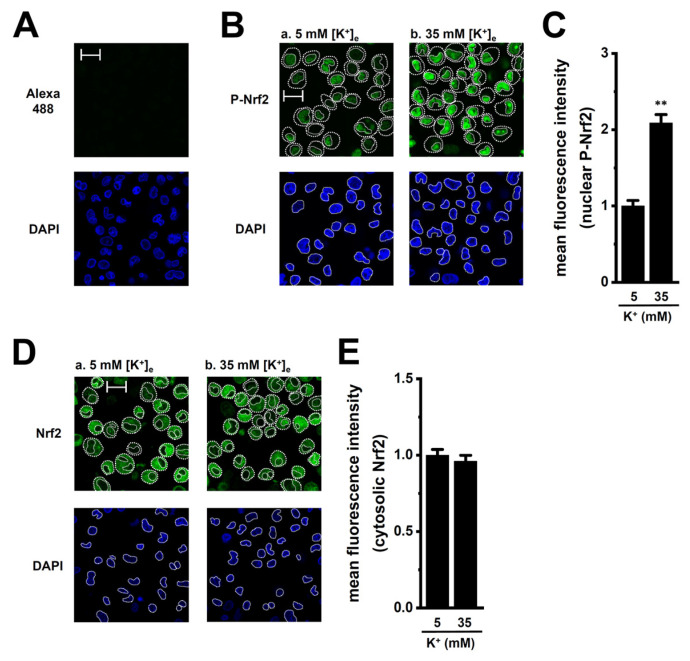
High [K^+^]_e_ promotes nuclear translocation of phosphorylated Nrf2 (P-Nrf2) in M_2_-MACs. (**A**): Confocal images showing Alexa Fluor 488 background fluorescence (upper panel) and DAPI (lower panel) in normal [K^+^]_e_ (5 mM)-treated cells. (**B**): Confocal images of P-Nrf2 (green) and DAPI (blue) in cells exposed to 5 mM (**a**) or 35 mM (**b**) [K^+^]_e_ for 2 h. Thick and thin dashed lines show the plasma membrane and nuclear boundary, respectively. (**C**): Quantification of mean intensity of nuclear P-Nrf2 fluorescence in both groups (*n* = 6). (**D**): Images of Nrf2 under the same conditions. (**E**): Quantification of mean intensity of cytosolic Nrf2 fluorescence (*n* = 6). Scale bars in (**A**,**B**,**D**) show 50 μm. **: *p* < 0.01 vs. normal [K^+^]_e_ (5 mM).

**Figure 5 ijms-26-07624-f005:**
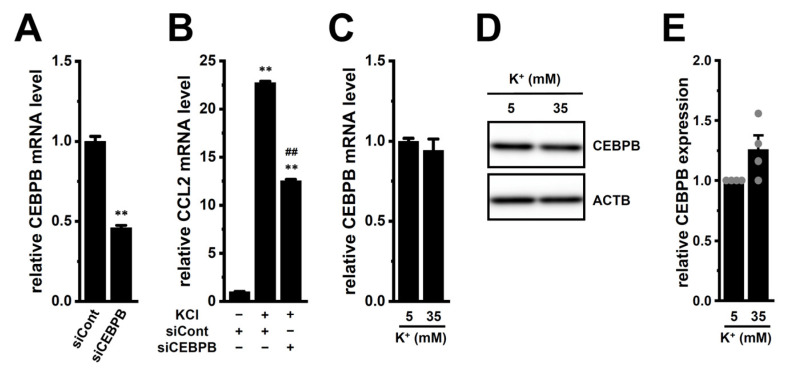
Effects of high [K^+^]_e_ on CEBPB expression and impact of siRNA-mediated CEBPB inhibition on CCL2 transcription in M_2_-MACs. (**A**): Knockdown efficiency of CEBPB siRNA (siCEBPB) after 48 h transfection in M_2_-MACs. mRNA levels are normalized to ACTB and shown relative to control siRNA (siCont) (*n* = 4). (**B**): CCL2 mRNA levels in M_2_-MACs transfected with siCont or siCEBPB and treated with 30 mM KCl for 24 h. Expression in siCont-transfected cells under normal K^+^ (5 mM) is set to 1.0 (*n* = 4). (**C**–**E**): CEBPB mRNA (**C**), protein expression (**D**), and densitometric quantification (**E**) in M_2_-MACs after exposure to normal or high [K^+^]_e_ (*n* = 4). **: *p* < 0.01 vs. −/+/− or siCont; ##: *p* < 0.01 vs. +/+/−.

**Figure 6 ijms-26-07624-f006:**
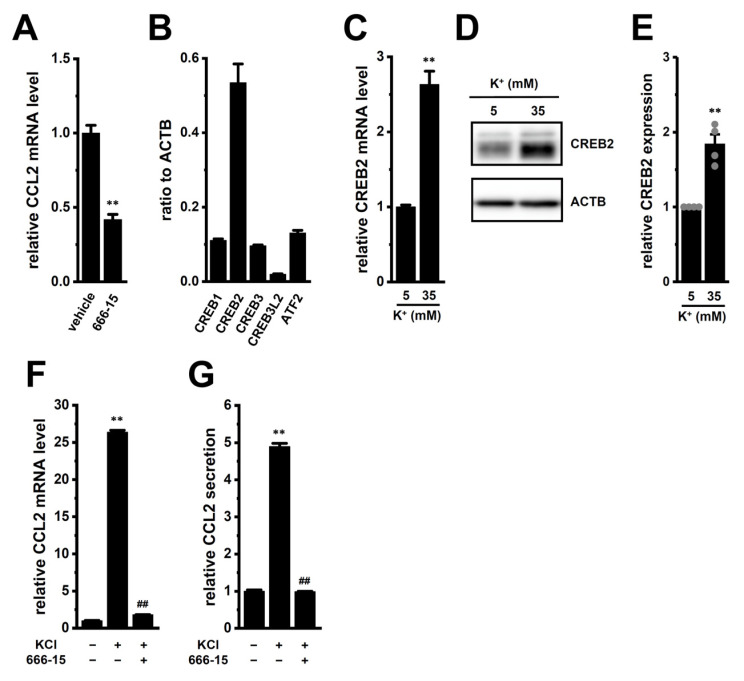
Effects of high [K^+^]_e_ on CREB isoforms and the role of CREB in CCL2 expression in M_2_-MACs. (**A**): CCL2 mRNA levels following 12 h treatment with vehicle or 1 μM 666-15 (pan-CREB inhibitor) (*n* = 4). (**B**): Expression profiles of CREB isoforms (CREB1, CREB2/ATF4, CREB3, CREB3L2, ATF2) in M_2_-MACs by real-time PCR (*n* = 4). Data are normalized to ACTB. (**C**–**E**): CREB2 mRNA (**C**), protein expression (**D**), and densitometric quantification (**E**) in M_2_-MACs under normal and high [K^+^]_e_ conditions (*n* = 4). (**F**,**G**): CCL2 mRNA (**F**) and secretion (**G**) after treatment with vehicle (−/−), 30 mM KCl (+/−), or KCl + 666-15 (+/+) (*n* = 4). **: *p* < 0.01 vs. vehicle control (−/−) or normal [K^+^]_e_; ##: *p* < 0.01 vs. +/−.

**Figure 7 ijms-26-07624-f007:**
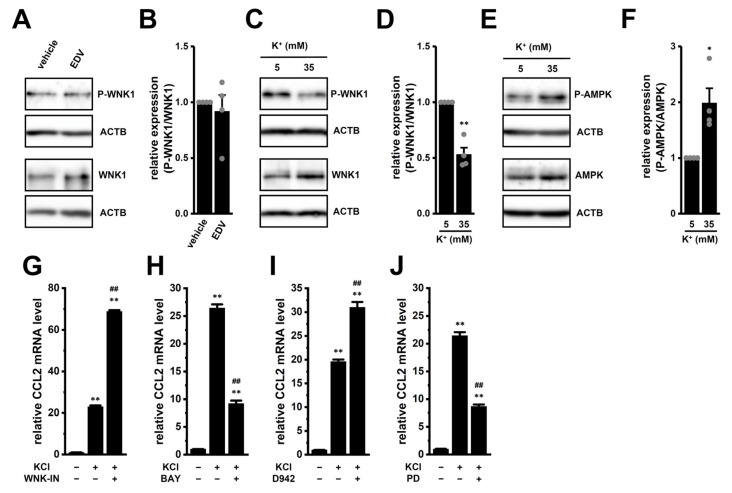
Effects of high [K^+^]_e_ and LRRC8A inhibition on WNK1 and AMPK phosphorylation in M_2_-MACs. (**A**,**B**): Western blot (**A**) and densitometry (**B**) of phospho-WNK1 (P-WNK1) and total WNK1 (WNK1) following treatment with 10 μM EDV. Specific band signals for P-WNK1 and WNK1 were observed at approximately 280 kDa. Expression in the vehicle control is set to 1.0 (*n* = 4). (**C**–**F**): P-WNK1 (**C**,**D**) and P-AMPK (**E**,**F**) expression under normal or high [K^+^]_e_. Specific band signals for P-AMPK and AMPK were observed at approximately 65 kDa. Expression in normal K^+^ is set to 1.0 (*n* = 4). (**G**–**J**): CCL2 mRNA levels after 12 h treatment with vehicle (−/−), 30 mM KCl (+/−), or KCl + 1 μM WNK-IN-11 (WNK-IN) (**G**), 1 μM BAY3827 (BAY) (**H**), 10 μM D942 (**I**), or 1 μM PD169316 (PD) (**J**) (+/+) (*n* = 4). *^,^**: *p* < 0.05, 0.01 vs. vehicle control, normal [K^+^]_e_, or −/−; ##: *p* < 0.01 vs. +/−.

**Figure 8 ijms-26-07624-f008:**
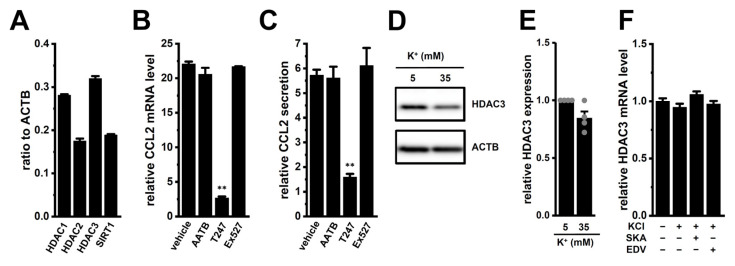
Role of HDAC3 in high [K^+^]_e_-induced CCL2 expression in M_2_-MACs. (**A**): Expression levels of HDAC1, HDAC2, HDAC3, and SIRT1 in M_2_-MACs by real-time PCR (*n* = 4). (**B**,**C**): CCL2 mRNA (**B**) and secretion (**C**) levels in M_2_-MACs treated with 10 μM AATB (HDAC1/2 inhibitor), 10 μM T247 (HDAC3 inhibitor), 10 μM Ex527 (SIRT1 inhibitor), or vehicle under high [K^+^]_e_ conditions (*n* = 4, 12 h). (**D**,**E**): Western blot (**D**) and densitometric quantification (**E**) of HDAC3 expression under high [K^+^]_e_ conditions (*n* = 4). (**F**): HDAC3 mRNA levels in M_2_-MACs treated with 30 mM KCl, KCl + SKA, or KCl + EDV for 12 h (*n* = 4). **: *p* < 0.01 vs. vehicle control.

**Figure 9 ijms-26-07624-f009:**
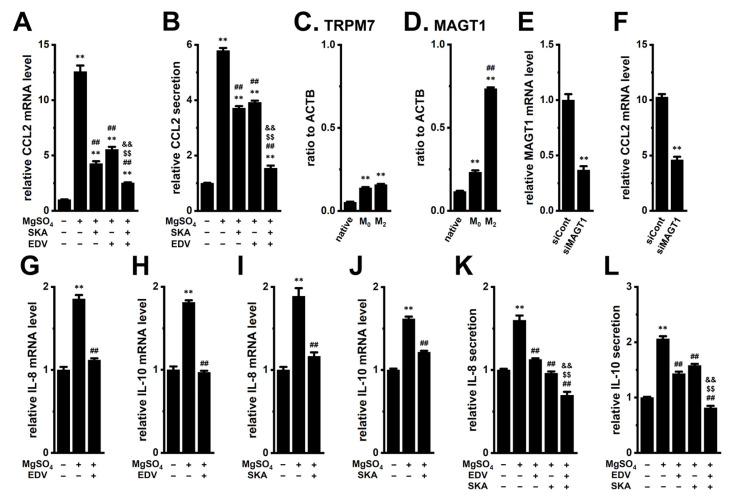
Suppressive effects of K_Ca_3.1 activation and LRRC8A inhibition on high [Mg^2+^]_e_-induced upregulation of CCL2, IL-8, and IL-10 in M_2_-MACs. (**A**,**B**): CCL2 expression (**A**) and secretion (**B**) in M_2_-MACs treated with vehicle (−/−/−), 19.6 mM MgSO_4_ (+/−/−), MgSO_4_ with 10 μM SKA (+/+/−), 10 μM EDV (+/−/+), or both for 12 and 24 h, respectively (*n* = 4). (**C**,**D**): Expression of TRPM7 (**C**) and MAGT1 (**D**) across native THP-1, M_0_-MACs, and M_2_-MACs (*n* = 4). (**E**): Knockdown efficiency of MAGT1 siRNA (siMAGT) after 48 h transfection in M_2_-MACs. mRNA levels are shown relative to control siRNA (siCont) (*n* = 4). (**F**): CCL2 expression in siCont and siMAGT1 groups (*n* = 4). (**G**–**J**): IL-8 (**G**,**I**) and IL-10 (**H**,**J**) expression in vehicle (−/−), MgSO_4_ (+/−), MgSO_4_ + EDV (+/+) (**G**,**H**), and MgSO_4_ + SKA (+/+) (**I**,**J**). (**K**,**L**): IL-8 and IL-10 secretion in the vehicle (−/−/−), MgSO_4_ (+/−/−), and MgSO_4_ + EDV (+/+/−), + SKA (+/−/+), or + both (+/+/+) groups (*n* = 4). **: *p* < 0.01 vs. vehicle control (−/− or −/−/−), native THP-1, or siCont; ##: *p* < 0.01 vs. +/−, +/−/−, or M_0_; $$: *p* < 0.01 vs. +/+/−; &&: *p* < 0.01 vs. +/−/+.

**Figure 10 ijms-26-07624-f010:**
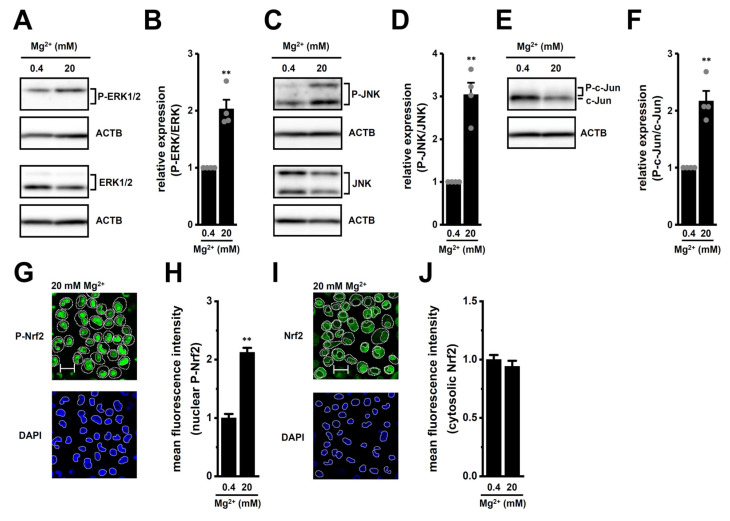
Effects of high [Mg^2+^]_e_ on ERK1/2 (P-ERK1/2), JNK (P-JNK), and c-Jun (P-c-Jun) phosphorylation and P-Nrf2 nuclear translocation in M_2_-MACs. (**A**–**F**): Western blot (**A**,**C**,**E**) and densitometric quantification (**B**,**D**,**F**) of phosphorylated and total ERK, JNK, and c-JUN under normal and high [Mg^2+^]_e_ (*n* = 4). (**G**–**J**): Confocal imaging and quantification of nuclear P-Nrf2 (**G**,**H**) and cytosolic Nrf2 (**I**,**J**) signals in high [Mg^2+^]_e_-exposed M_2_-MACs (*n* = 6). Scale bars in (**G**,**I**) show 50 μm. **: *p* < 0.01 vs. normal [Mg^2+^]_e_.

**Figure 11 ijms-26-07624-f011:**
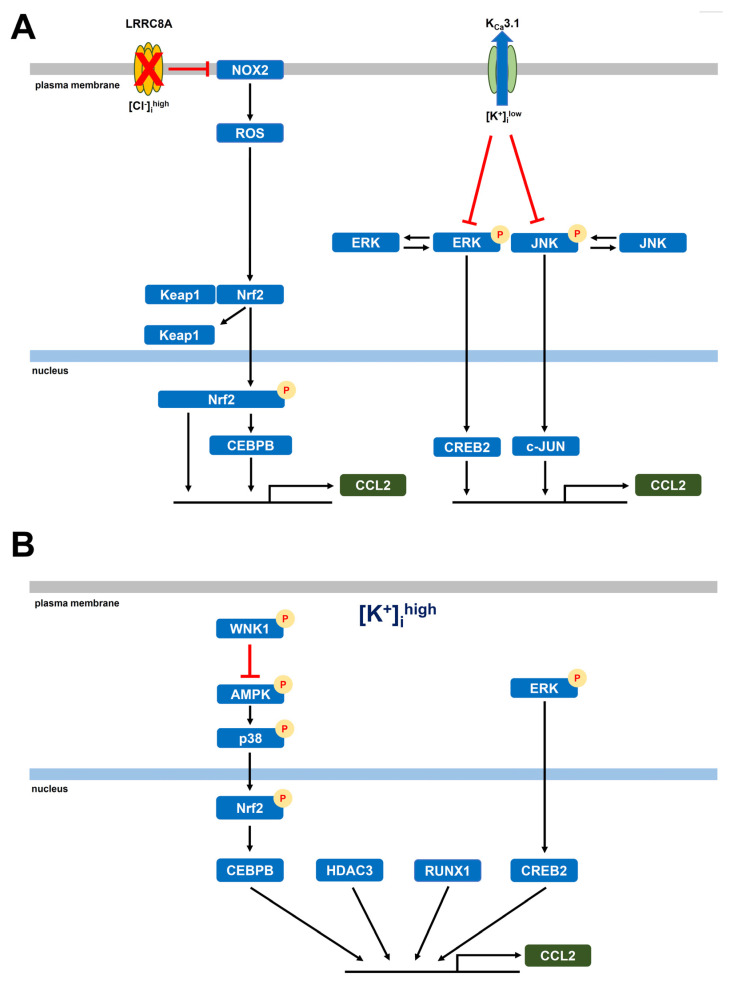
Schematic diagram of the intracellular signaling pathways involved in K_Ca_3.1-/LRRC8A-mediated (**A**) and high [K^+^]_e_-exposed (**B**) CCL2 transcription in M_2_-MACs. (**A**): Activation of K_Ca_3.1 inactivates the downstream of the ERK–CREB2 and JNK–c-Jun axes. Inhibition of LRRC8A inactivates the downstream of the NOX2–Nrf2–CEBPB axis. (**B**): High [K^+^]_i_ exposure activates the WNK1–AMPK–Nrf2–CEBPB, HDAC3, RUNX1, and ERK–CREB2 transcriptional axes in M_2_-MACs.

## Data Availability

The original contributions presented in the study are included in the article/Appendix A. Further inquiries can be directed to the corresponding author.

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
