# Peer review of "Transcriptional Repression of CCL2 by K_Ca_3.1 K^+^ Channel Activation and LRRC8A Anion Channel Inhibition in THP-1-Differentiated M_2_ Macrophages"

_ijms, 2025, doi:10.3390/ijms26157624_

Round 1

Reviewer 1 Report

Comments and Suggestions for Authors

This manuscript explores the transcriptional regulation of the chemokine CCL2 in M2-polarized macrophages via modulation of ion channels KCa3.1 and LRRC8A, using THP-1-derived macrophage models. This study provides mechanistic insights into ERK–CREB, JNK–c-Jun, and NOX2–Nrf2–CEBPB signaling pathways and their role in modulating CCL2 expression under tumor-mimicking conditions. While the findings are of interest and potentially relevant to tumor immunotherapy, several experimental and presentation-related issues require attention.

  • The number of biological replicates (n) should be clearly stated for each experiment, particularly for the Western blots. Quantification without visible replicate data can mask variability or bias.

  • Bar graphs summarizing Western blot results should display individual data points where possible, as solid bars alone may conceal skewed distributions or outliers.

  • Statistical annotations (e.g., p-values or asterisk indicators) on graphs are too small to be legible and should be resized for clarity.

  • The authors should confirm that pharmacological inhibition of ERK, JNK, NOX2, and Nrf2 did not compromise cell viability at the time RNA was collected for qPCR. A viability assay is strongly recommended to support this.

  • In Figure 5/6D, loading controls (e.g., β-actin or GAPDH) are missing and should be included to validate equal protein loading across samples.

  • The authors should provide quantification of Western blots from at least three independent biological replicates and include corresponding statistical analyses.

  • For all panels in Figure 5, visual representation of replicates and consistent presentation of error bars are necessary to support reproducibility.

  • Given the successful derivation of M2 macrophages, the authors are encouraged to include a comparative analysis with M1 macrophages, not only naïve or undifferentiated cells, to better contextualize the specificity of observed regulatory pathways.

  • The manuscript's title and subheadings (lines 5–7) appear fragmented. Consider unifying them into a single coherent title that clearly reflects the central theme of the study.

  • The author list and affiliation section need formatting corrections to meet standard journal conventions. Email formatting and author designation (†, *) should be reviewed for consistency.

  • The abstract effectively communicates the study’s scope, but it would benefit from clearer background context on the role of M2 macrophages in the tumor microenvironment.

  • The abstract should clearly define the experimental model (e.g., THP-1-derived macrophages) at the start to help readers understand the study's system.

  • The pathways identified (ERK–CREB, JNK–c-Jun, NOX2–Nrf2–CEBPB, and WNK1–AMPK) are mechanistically interesting; however, the abstract could be more concise in how these are presented to emphasize their distinct roles in CCL2 regulation.

  • The final sentence of the abstract should explicitly state how the proposed modulation of ion channels could inform therapeutic strategies targeting tumor-associated macrophages.

Author Response

Responses to Reviewer 1

We would like to thank the reviewer for his/her valuable comments. We have attended to all the points raised by the reviewers. Each comment is highlighted below with our response underneath.

  1. The number of biological replicates (n) should be clearly stated for each experiment, particularly for the Western blots. Quantification without visible replicate data can mask variability or bias.

We agree with the reviewer’s comment and have included the number of biological replicates (n) for each experiment. To address the following comment as well, we have plotted the individual data points on the bar graphs for the Western blot results.

  1. Bar graphs summarizing Western blot results should display individual data points where possible, as solid bars alone may conceal skewed distributions or outliers.

According to the reviewer’s comment, we have added individual data points to the bar graphs of the Western blot results, in addition to the mean values and error bars.

  1. Statistical annotations (e.g., p-values or asterisk indicators) on graphs are too small to be legible and should be resized for clarity.

According to the reviewer’s indication, we enlarged all asterisk indicators on graphs.

  1. The authors should confirm that pharmacological inhibition of ERK, JNK, NOX2, and Nrf2 did not compromise cell viability at the time RNA was collected for qPCR. A viability assay is strongly recommended to support this.

We confirmed that the compounds used in this study did not significantly affect cell viability. The corresponding data have been included in Supplementary Figure S3. Thank you for your careful reading and adequate suggestions.

  1. In Figure 5/6D, loading controls (e.g., β-actin or GAPDH) are missing and should be included to validate equal protein loading across samples.

In the real-time PCR and Western blot experiments, β-actin (‘ACTB’) was used as an internal control to validate equal protein loading. See Sections 4.3 and 4.4.

  1. The authors should provide quantification of Western blots from at least three independent biological replicates and include corresponding statistical analyses.

Protein samples were extracted from at least four independent biological replicates, each with technical replicates, and the summarized results in Western blots are not derived from quadruplicate samples. In addition, we used a paired t-test for the Western blot experiments. We have added these descriptions to Section 4.4.

  1. For all panels in Figure 5, visual representation of replicates and consistent presentation of error bars are necessary to support reproducibility.

According to the reviewer’s comment, we amended Figure 5E (similarly to Comments 2 and 6).

  1. Given the successful derivation of M2 macrophages, the authors are encouraged to include a comparative analysis with M1 macrophages, not only naïve or undifferentiated cells, to better contextualize the specificity of observed regulatory pathways.

According to the reviewer’s comment, we examined the comparative studies using THP-1-differentiated M1-like macrophages (M1-MACs) stimulated by lipopolysaccharide (LPS) and interferon (IFN)-γ. In M1-MACs, IL-1β was abundantly expressed compared with M2-MACs (Fig. C), and M2-markers, CCL5 and CCL22 were less abundantly expressed (not shown). The expression levels of KCa3.1 and LRRC8A in M1-MACs were lower than those in M2-MACs (Fig. A, B). CCL2 is also produced in M1 macrophages; however, activation of KCa3.1 with SKA121 (SKA) and inhibition of LRRC8A with endovion (EDV) did not affect the CCL2 expression level in M1-MACs (Fig. D). Similar to M2-MACs, exposure to high [K+]e increased CCL2 mRNA level (Fig. E), and CCL2 transcription was suppressed by the inhibition of NOX2, Nrf2, ERK, and CREB (Fig. F). Different from M2-MACs, CCL2 was not affected by the inhibition of JNK in M1-MACs (Fig. F). As shown in Figure D, KCa3.1 and LRRC8A did not affect CCL2 expression in M1-MACs. Therefore, we did not add the descriptions of M1-MACs in the present study. We would like to clarify the role of ion channels in M1 markers such as IL-1β in the future study.

Figure. Effects of the KCa3.1 activation, LRRC8A inhibition, high [K+]e exposure, and inhibition of NOX2, Nrf2, ERK, JNK, and CREB on CCL2 mRNA expression in THP-1-differentiated M1 macrophages (M1-MACs). A, B: Real-time PCR examination of KCa3.1 (A) and LRRC8A (B) mRNA expression in native THP-1, M0-MACs, and M1-MACs. Data were normalized to ACTB (n = 4). C: Real-time PCR examination of IL-1B mRNA expression in native THP-1, M0-MACs, M1-MAC, and M2-MACs (n = 4). D, E: CCL2 mRNA expression following treatment with vehicle (/), 10 μM SKA121 (SKA) (+/), and 10 μM endovion (EDV) (‒/+) (D) and exposure to high [K+]e (35 mM) (E) for 12-hr in M1-MACs. Expression levels are shown relative to vehicle control (set as 1.0, n =4). F: CCL2 mRNA expression following the treatment with a NOX2 inhibitor, GSK2795039 (GSK, 10 μM), an Nrf2 inhibitor, ML385 (5 μM), an ERK inhibitor, SCH772984 (SCH, 1 μM), a JNK inhibitor, SP600125 (SP, 1 μM), and a pan-CREB inhibitor, 666-15 (1 μM) (n = 4). **P < 0.01 vs. native THP-1, normal [K+]e (5 mM), and vehicle control.    

  1. The manuscript's title and subheadings (lines 5–7) appear fragmented. Consider unifying them into a single coherent title that clearly reflects the central theme of the study.

We agreed with the reviewer’s suggestion and have reconsidered the title accordingly (see the revised title).

  1. The author list and affiliation section need formatting corrections to meet standard journal conventions. Email formatting and author designation (†, *) should be reviewed for consistency.

The formatting and notation were revised following the IJMS submission guidelines.

  1. The abstract effectively communicates the study’s scope, but it would benefit from clearer background context on the role of M2 macrophages in the tumor microenvironment.

According to the reviewer’s suggestion, we described the significance of M2-polarized TAM clearly in the first paragraph of the ‘Introduction’ section (Section 1).

  1. The abstract should clearly define the experimental model (e.g., THP-1-derived macrophages) at the start to help readers understand the study's system.

According to the reviewer’s indication, we defined the experimental model using THP-1 cells at the beginning of the ‘Abstract’.

  1. The pathways identified (ERK–CREB, JNK–c-Jun, NOX2–Nrf2–CEBPB, and WNK1–AMPK) are mechanistically interesting; however, the abstract could be more concise in how these are presented to emphasize their distinct roles in CCL2 regulation.

In the original version, we described that ‘CCL2 expression is regulated through the ERK‒CREB2, JNK, and Nrf2 transcriptional axes in various types of cells, including macrophages’ in the ‘Results’ section (Section 3.2). In agreement with the reviewer’s comment, we added the description of CCL2-related pathways in the ‘Abstract’ section.

  1. The final sentence of the abstract should explicitly state how the proposed modulation of ion channels could inform therapeutic strategies targeting tumor-associated macrophages.

According to the reviewer’s comment, we described how the proposed modulation of ion channels could inform therapeutic strategies targeting tumor-associated macrophages in the final sentence of the ‘Abstract’ section.

Reviewer 2 Report

Comments and Suggestions for Authors

Review on "Transcriptional  Repression  of  CCL2  by  KCa3.1  K+  Channel Activation  and  LRRC8A  Cl-  Channel  Inhibition  in  THP1-Derived M2 Macrophages" by Matsui et al.

In this report, the authors investigated regulatory mechanisms of CCL2 in macrophage-like THP-1 (differentiated by PMA) based on their theory that high CCL2 expression is a crucial step for immune suppression by M2 macrophages in the tumor microenvironment (TME). The authors found that high intracellular K+ and Mg2+ concentrations can upregulate CCL2 and consequently increase mRNA for IL8 and IL10, which have been associated with low T-cell infiltration and an increase in MDSCs in the TME. The authors found that either activating K+ efflux transporter Kca3.1 or inhibiting K+ importer LRRC8A resulted in the suppression of CCL2. This study further explored responsible signal transduction pathways using inhibitors of ERK (SCH772984), JNK (SP600125), NOX2 (GSK2795039), and Nrf2 (ML385) to conclude that both the ERK-CREB2 and AMPK-Nrf2 pathways contribute to the CCL2 upregulation after K+ influx.

Most experiments were well-designed and conducted using Western blot and qRT-PCR after treating THP-1 (differentiated by PMA then by IL4 + IL13 with or without siRNA when used). ELISA was employed to evaluate cytokine productions by the cells. The results were reasonably interpreted. A turbidimetric assay was utilized to assess [K+]i concentration.

The primary limitation of the study lies in the absence of in vivo experiments, which hinders the assessment of the biological relevance of the findings. The majority of the figures were based on THP-1 cells, making it challenging to generalize the observations. This is particularly significant considering that studies involving infiltrating microglia/macrophages in glioma have demonstrated high expression of Kca3.1. Furthermore, inhibition of Kca3.1 was shown to suppress tumor growth (PMID 27054329). These observations are inconsistent with the primary conclusion of the present study. Thus, it is plausible that the findings obtained using THP-1 cells may be specific and may not be applicable to broader populations.

Other comments

What types of cancer do the authors focus on? In glioma-focused studies, inhibition of KCa3.1 in infiltrating microglia/macrophages has been shown to suppress tumor growth, which contrasts with the conclusions of the present study. If the authors intend to explore the therapeutic use of SKA (a KCa3.1 activator) and EDV (an LRRC8A inhibitor), they should discuss the specific cancer types under consideration (likely ones that differ from gliomas).

As noted above, there are a few previous studies with findings that differ from the present results, which the authors may wish to acknowledge and discuss for clarity.

 - PMID 27054329 and 36589283. These studies must be cited to discuss the differences between their findings and this study and possible reasons why the opposite effect was observed.

- PMID 35689211: KCNN4 may weaken anti-tumor immune response via raising Tregs and diminishing resting mast cells in clear cell renal cell carcinoma, 2022

- PMID 34295732: KCNN4 is a potential prognostic marker and critical factor affecting the immune status of the tumor microenvironment in kidney renal clear cell carcinoma, 2021

The immunocytochemistry shown in Fig. 4 would benefit from a more quantitative approach. Rather than staining separately with Alexa Fluor 488 for both targets, cells should be co-stained simultaneously with anti-pNrf2 and anti-Nrf2 antibodies. This would allow for calculation of the pNrf2-to-Nrf2 ratio within the same cells. Such co-staining is straightforward using antibodies conjugated with different fluorophores (e.g., Alexa Fluor 488 and Alexa Fluor 568 or 647).

Author Response

Responses to Reviewer 2

 We would like to thank the reviewer for his/her valuable comments. We have attended to all the points raised by the reviewers. Each comment is highlighted below with our response underneath.

  1. What types of cancer do the authors focus on? In glioma-focused studies, inhibition of KCa3.1 in infiltrating microglia/macrophages has been shown to suppress tumor growth, which contrasts with the conclusions of the present study. If the authors intend to explore the therapeutic use of SKA (a KCa3.1 activator) and EDV (an LRRC8A inhibitor), they should discuss the specific cancer types under consideration (likely ones that differ from gliomas).

As the reviewer pointed out, in cancer types that functionally express KCa3.1, activation of KCa3.1 may potentially promote cellular proliferation and invasiveness. However, in solid tumors, non-malignant cells are often reported to comprise approximately 30–70% of the total tumor mass. While the proportion of non-cancerous cells varies greatly depending on the cancer type, stage, and so on, immunosuppressive cells, including TAMs, typically account for about 10–30%. Therefore, even in KCa3.1-positive solid tumors, if immunosuppressive cells such as Tregs, TAMs, or MDSCs represent a significant portion of the tumor, activating KCa3.1 may suppress the function of these immunosuppressive cells, restore immune surveillance, and thereby inhibit tumor growth. Indeed, in our unpublished data, administration of a KCa3.1 inhibitor in mice transplanted with KCa3.1-expressing osteosarcoma cells resulted in increased tumor weight, suggesting a possible tumor-promoting effect of KCa3.1 inhibition in this context. We are particularly interested in drug discovery targeting KCa3.1 in cancers such as breast and prostate cancer. However, for the reasons described above, we would prefer not to explicitly discuss whether our findings are inapplicable to KCa3.1-expressing cancers, nor to restrict the scope to specific tumor types.

We added the descriptions above, excluding our unpublished results (Page 16, line 512-521). We appreciate the reviewer’s careful reading and adequate suggestions.

  1. As noted above, there are a few previous studies with findings that differ from the present results, which the authors may wish to acknowledge and discuss for clarity.

 - PMID 27054329 and 36589283. These studies must be cited to discuss the differences between their findings and this study and possible reasons why the opposite effect was observed.

- PMID 35689211: KCNN4 may weaken anti-tumor immune response via raising Tregs and diminishing resting mast cells in clear cell renal cell carcinoma, 2022

- PMID 34295732: KCNN4 is a potential prognostic marker and critical factor affecting the immune status of the tumor microenvironment in kidney renal clear cell carcinoma, 2021

According to the reviewer’s suggestion, the following descriptions were added in the ‘Discussion’ section (Section 5). (Page 16, line 521-531)

Several studies, which report findings that contrast with those of the present study, have demonstrated that KCa3.1 inhibitors can attenuate tumor progression and improve clinical outcomes by modulating the phenotypic and functional properties of immune cells within the TME. In murine models of glioma, pharmacological inhibition of KCa3.1 has been shown to reduce tumor burden by promoting a phenotypic shift of TAMs from a pro-tumorigenic M2 to an anti-tumorigenic M1 phenotype [Grimaldi et al., 2016; Massenzio et al., 2022]. Similarly, in renal clear cell carcinoma, transcriptomic analyses of tumor-infiltrating immune cells have revealed a negative correlation between KCa3.1 expression levels in tumor burden and patient prognosis [Chen et al., 2021]. These findings underscore the need for further investigation to delineate the specific cancer types and stages in which KCa3.1 activators or LRRC8A inhibitors may exert therapeutic efficacy.

<references>

Grimaldi A, D'Alessandro G, Golia MT, Grössinger EM, Di Angelantonio S, Ragozzino D, Santoro A, Esposito V, Wulff H, Catalano M, Limatola C. KCa3.1 inhibition switches the phenotype of glioma-infiltrating microglia/macrophages. Cell Death Dis. 2016;7:e2174. doi: 10.1038/cddis.2016.73.

Massenzio F, Cambiaghi M, Marchiotto F, Boriero D, Limatola C, D'Alessandro G, Buffelli M. In vivo morphological alterations of TAMs during KCa3.1 inhibition-by using in vivo two-photon time-lapse technology. Front Cell Neurosci. 2022;16:1002487. doi: 10.3389/fncel.2022.1002487.

Chen S, Wang C, Su X, Dai X, Li S, Mo Z. KCNN4 is a potential prognostic marker and critical factor affecting the immune status of the tumor microenvironment in kidney renal clear cell carcinoma. Transl Androl Urol. 2021;10:2454-2470. doi: 10.21037/tau-21-332.

  1. The immunocytochemistry shown in Fig. 4 would benefit from a more quantitative approach. Rather than staining separately with Alexa Fluor 488 for both targets, cells should be co-stained simultaneously with anti-pNrf2 and anti-Nrf2 antibodies. This would allow for calculation of the pNrf2-to-Nrf2 ratio within the same cells. Such co-staining is straightforward using antibodies conjugated with different fluorophores (e.g., Alexa Fluor 488 and Alexa Fluor 568 or 647).

Corresponding to the reviewer’s opinion on Fig. 4, we also distinguish that co-staining simultaneously by anti-P-Nrf2 and anti-Nrf2 conjugated with two different Alexa fluorophores is valuable for a ‘more quantitative’ approach. In our previous study (Matsui M. et al., Int. J. Mol. Sci., 2024 (PMID: 39273558), we estimated the P-Nrf2 levels as the percentages of P-Nrf2-positive cells. However, this approach is not major. Therefore, we calculated the mean fluorescence intensity of P-Nrf2 signals in the nuclei in this study. Of course, cell immunostaining, imaging acquisition, and quantitative analyses were performed independently by separate investigators to minimize potential bias. We added this description in Section 4.7.

We also noticed the errors in the description of the data acquisition and amended it as follows,

For each batch (n =1), the fluorescence intensities were obtained from at least 6 frame images, including more than 30 cells.

I appreciate the reviewer’s professional suggestion and careful reading of our manuscript. We would like to perform simultaneous imaging of P-Nrf2 and Nrf2 in our future studies.

Reviewer 3 Report

Comments and Suggestions for Authors

The study by Miki Matsui et al. investigates the role of Kca3.1 K+ channels and LRRC8A Cl- channels in regulating CCL2 expression in THP-1-derived M2 macrophages, mimicking the tumor microenvironment (TME). The findings suggest that targeting these channels could be a novel strategy for cancer immunotherapy by inhibiting TAM recruitment. The research presents interesting findings. However, some sections should be revised.

Overall assessment:

The study explores a relatively under-investigated area and is novel. Also, the finding that both K+ and Mg2+ can upregulate CCL2 is particularly interesting. The authors evaluated the signaling pathways involved in CCL2 regulation by Kca3.1 and LRRC8A, linking them to ERK-CREB2, JNK-c-JUN, and NOX2-Nrf2-CEBPB pathways, which is also interesting. However, the entire study is conducted using an in vitro THP-1 derived M2 macrophage model, which is the main limitation.

The absence of ex vivo human TAMs or in vivo models makes the direct translation of these findings speculative. Also, the off-target effects of some pharmacological agents, especially at higher concentrations or prolonged exposure, cannot be entirely ruled out without further validation (such as using genetic knockouts/knockdowns for all targets). The study focuses solely on M2 macrophages. While TAMs are heterogeneous, and investigating the effects on other macrophage phenotypes or during polarization is important. Accordingly, MAJOR REVISIONS are suggested before publication.

Section-by-Section Review:

Abstract

I suggest rephrasing the abstract by clearly state the distinct mechanisms for Kca3.1 and LRRC8A earlier, and then mention the high [K+]e effect and its reversal as a key finding.  The concluding sentence is strong but could be more impactful if the key takeaway from the entire study is crystal clear.

  1. Introduction

(A) The introduction provides good background on TAMs, CCL2, K+ channels, and LRRC8A. However, the reason for investigating both Kca3.1 and LRRC8A in this specific context (CCL2 regulation) should be stated as a core hypothesis.

(B) The connection between their previous work on IL-10/IL-8 and the current focus on CCL2 is mentioned, but the unique importance of CCL2 in the TME, justifying a separate study, should be discussed. Clearly articulate why these two channels are hypothesized to regulate CCL2, beyond their known roles in IL-10/IL-8. Discuss the specific gap in the literature that this study aims to fill regarding CCL2 regulation by these ion channels and ionic imbalances in the TME.

  1. Results

(A) "n=4 for each" repetition: This phrase is extremely repetitive. It should be stated once in the Materials and Methods section for each type of experiment ("All experiments were performed with at least four independent biological replicates unless otherwise stated.") and then only specifically mentioned if an N-value deviates.

(B) For figures 1A and 1B, it would be helpful to include a control, like untreated THP-1 cells, to clearly show the differentiation effect on CCL22 and CCL2 expression. This is partially addressed by "native THP-1 cells", but further emphasis on the fold change or absolute expression difference between native and M2-MACs would be beneficial.

(C) The finding that "exposure to 20 mM [Mg2+] by the addition of 19.6 mM MgSO4 significantly increased CCL2 transcript levels by approximately 10-fold" is very significant but is almost presented as an afterthought in this section, which is primarily focused on K+. Create a dedicated subsection for the Mg2+ findings to give them the appropriate emphasis.

(D) When introducing high [K+]e, specify how "high" is defined (physiological range vs. TME conditions). The 35 mM is stated, which is good, but a brief note on its relevance to the TME (pathological levels) is helpful.

(E) Consider including a simplified diagram in the results or discussion to visually represent the proposed signaling pathways and their convergence/divergence for CCL2 and IL-10.

(F) Acknowledge the lower siRNA efficiency as a minor limitation and discuss how future studies could use CRISPR/Cas9 or other methods for more complete gene knockout if feasible.

(G) A more detailed statement about the independent action of LRRC8A inhibition from the WNK1-AMPK-Nrf2 axis at the beginning of this subsection would aid understanding.

(H) (MAJOR) Since HDAC3 activity is mentioned, directly assessing HDAC3 activity (histone deacetylation assays) can provide stronger evidence than just using inhibitors. If it is not possible or feasible to conduct, discuss how future studies could measure HDAC3 activity to confirm its functional involvement.

(I) (MAJOR) While MAGT1 was upregulated, direct evidence of its role in Mg2+ transport and subsequent CCL2 regulation (MAGT1 knockdown) is not studied. Include experiments that directly test the involvement of MAGT1, such as siRNA knockdown of MAGT1, to confirm its role in Mg2+-induced CCL2 upregulation.

(J) In figure 1, the y-axis labels "ratio to ACTB" (A, B) and "relative CCL22 mRNA level" / "relative CCL2 mRNA level" (C, D, F, G) are inconsistent. It would be clearer to use a unified "Relative mRNA expression (normalized to ACTB)" or "Fold change vs. control" for all.

(K) In figure 2,  the notation in the x-axis "KCl +/-/-", "+/+/-" etc. is somewhat difficult to understand. A legend defining these combinations ("-": vehicle, "+": 30mM KCl, "SKA": SKA121, "EDV": EDV) within the figure caption or panel is necessary. Also, consider showing absolute intracellular K+ concentrations, or at least a standard curve, to provide more quantitative context.

In figure 5 and figure 6, label the lanes in D sections for K+ concentration. Also, it seems like that the ACTB expression as a HKG is not consistent in all experiments. Clarify.

  1. Discussion

(A) The discussion acknowledges the limitation of using THP-1 cells, but it doesn't sufficiently discuss how this model limits the findings or what specific differences might exist between THP-1 derived M2-MACs and in vivo TAMs. For example, TAMs are highly heterogeneous and influenced by various factors in the TME not replicated in this model. Enrich this section.

(B) Several connections, such as the potential involvement of RUNX1-HDAC3/CREB2/CEBPB axes  or specific miRNAs, are introduced based on existing literature but not directly investigated. While good for future research, they should be presented as hypotheses more clearly.

(C) Propose concrete next steps using primary human TAMs, co-culture systems with tumor cells, or relevant in vivo animal models to validate the findings. Discuss the complexities of TAM heterogeneity and how this in vitro model might misrepresent it.

  1. Materials and Methods

(A) The timing of drug applications (48- or 60-hr after IL-4/IL-13 incubation ) should be more justified in relation to the differentiation state of the M2-MACs. Are they fully differentiated and stable by this point?

(B) For the confocal imaging, clarify whether 'n' refers to biological replicates. If not, multiple independent experiments should be performed, with results from each replicate quantified and analyzed. (MAJOR)

(C) Include full supplier names and locations for all reagents and kits for maximum reproducibility.

Also, specify the software versions used for densitometry and real-time PCR analysis.

Comments on the Quality of English Language

Minor editing is needed for the language.

Author Response

Responses to Reviewer 3

 We would like to thank the reviewer for his/her valuable comments. We have attended to all the points raised by the reviewers. Each comment is highlighted below with our response underneath.

 Abstract

  1. I suggest rephrasing the abstract by clearly state the distinct mechanisms for KCa3.1 and LRRC8A earlier, and then mention the high [K+]e effect and its reversal as a key finding. 

According to the reviewer’s suggestion, we amended the ‘Abstract’ section.

  1. The concluding sentence is strong but could be more impactful if the key takeaway from the entire study is crystal clear.

According to the reviewer’s comment, we rearranged the ‘Abstract’ section.

Introduction

  1. The introduction provides good background on TAMs, CCL2, K+ channels, and LRRC8A. However, the reason for investigating both KCa3.1 and LRRC8A in this specific context (CCL2 regulation) should be stated as a core hypothesis.

According to the reviewer’s suggestion, we amended the final paragraph of the ‘Introduction’ section.

  1. The connection between their previous work on IL-10/IL-8 and the current focus on CCL2 is mentioned, but the unique importance of CCL2 in the TME, justifying a separate study, should be discussed (2-1). Clearly articulate why these two channels are hypothesized to regulate CCL2, beyond their known roles in IL-10/IL-8 (2-2). Discuss the specific gap in the literature that this study aims to fill regarding CCL2 regulation by these ion channels and ionic imbalances in the TME (2-3).

2-1. According to the reviewer’s suggestion that the authors should mention ‘the unique importance of CCL2 in the TME’, we amended the first paragraph of the ‘Introduction’ section.

2-2. In the original version, we described the reason why KCa3.1 and LRRC8A are hypothesized to regulate CCL2, beyond their known roles in IL-10/IL-8, in the ‘Results’ Section 2.3. According to the reviewer’s comment, we added these descriptions to the final paragraph of the ‘Introduction’ section.

2-3. According to the reviewer’s comment, we further added the following descriptions to the final paragraph of the ‘Introduction’ section.

The TME is characterized by altered ion concentrations, such as elevated [K⁺]ₑ [9]. Ionic shifts through ion channels and transporters can influence the behavior of M2-polarized TAMs by modulating intracellular signaling pathways. Ionic remodeling within the TME may contribute to CCL2, promoting the recruitment of monocytes and the maintenance of a pro-tumoral microenvironment.

Results

  1. "n=4 for each" repetition: This phrase is extremely repetitive. It should be stated once in the Materials and Methods section for each type of experiment ("All experiments were performed with at least four independent biological replicates unless otherwise stated.") and then only specifically mentioned if an N-value deviates.

According to the reviewer’s comment, we removed "n = 4 for each" in the main text and added the description that “All experiments were performed with at least four independent biological replicates unless otherwise stated” in Section 4.8.

  1. For figures 1A and 1B, it would be helpful to include a control, like untreated THP-1 cells, to clearly show the differentiation effect on CCL22 and CCL2 expression. This is partially addressed by "native THP-1 cells", but further emphasis on the fold change or absolute expression difference between native and M2-MACs would be beneficial.

We would like to express the target gene expression as a ratio to actin when using it for the first time. According to the reviewer’s comment, we added the fold change between native and M2-MACs in the text (Section 2.1). (Page 3, lines 93-94)

  1. The finding that "exposure to 20 mM [Mg2+] by the addition of 19.6 mM MgSO4 significantly increased CCL2 transcript levels by approximately 10-fold" is very significant but is almost presented as an afterthought in this section, which is primarily focused on K+. Create a dedicated subsection for the Mg2+ findings to give them the appropriate emphasis.

According to the reviewer’s suggestion, we created a subsection for the Mg2+ findings. To emphasize these findings, Supplementary Figure S2 was moved to the main text (Fig. 10).

  1. When introducing high [K+]e, specify how "high" is defined (physiological range vs. TME conditions). The 35 mM is stated, which is good, but a brief note on its relevance to the TME (pathological levels) is helpful.

Tan et al. (2020) have shown that K+ levels are elevated in the TME, with an average concentration of approximately 29 mM. Eil et al. also reported that the K+ concentration in tumor interstitial fluid was elevated up to 30 - 80 mM (average 40 mM) [9]. We added this description in Section 2.2. (Page 3, line 113-116)

Tan, J.W.Y.; Folz, J.; Kopelman, R.; Wang, X. In vivo photoacoustic potassium imaging of the tumor microenvironment. Biomed. Opt. Express 2020, 11, 3507-3522. doi: 10.1364/BE.393370.

  1. Consider including a simplified diagram in the results or discussion to visually represent the proposed signaling pathways and their convergence/divergence for CCL2 and IL-10.

According to the reviewer’s suggestion, we added the schematic diagram, which was submitted as a graphic summary in the original manuscript, to the main text (Figure 11).

  1. Acknowledge the lower siRNA efficiency as a minor limitation and discuss how future studies could use CRISPR/Cas9 or other methods for more complete gene knockout if feasible.

According to the reviewer’s indication, we added the following descriptions in the ‘Discussion’ section. (Page 16, line 502-506)

In the present study, the knockdown efficiency of the siRNAs was relatively low. Future studies could employ more robust gene-editing approaches, such as CRISPR/Cas9-mediated knockout, to achieve more complete and stable gene disruption. This would enable a more comprehensive understanding of the target molecule’s function under the experimental conditions.

  1. A more detailed statement about the independent action of LRRC8A inhibition from the WNK1‒AMPK‒Nrf2 axis at the beginning of this subsection would aid understanding.

We described the independent action of LRRC8A inhibition from the WNK1AMPKNrf2 axis in the ‘Results’ section (Section 2.5). (Page 5, line 212-213) It is difficult to move this sentence to the beginning of this subsection. To emphasize’ the independent action of LRRC8A inhibition from the WNK1AMPKNrf2 axis’, we described it again in the first paragraph, main finding (2) of the ‘Discussion’ section. (Page 14, line 412-413)

  1. (MAJOR) Since HDAC3 activity is mentioned, directly assessing HDAC3 activity (histone deacetylation assays) can provide stronger evidence than just using inhibitors. If it is not possible or feasible to conduct, discuss how future studies could measure HDAC3 activity to confirm its functional involvement.

We agree with the reviewer’s comment. The effects of pharmacological inhibition and siRNA-mediated knockdown of HDACs are ‘indirect’ evidence. If we describe the involvement of HDACs in the epigenetic modification of CCL2, it is necessary to ‘directly’ measure histone deacetylation status by ChIP assay or HDAC enzymatic activity. We added the following descriptions in the ‘Discussion’ section. (Page 14, line 426-429) We appreciate the reviewer’s adequate advice for a more precise interpretation of our results.

The effect of pharmacological inhibition of HDACs is indirect evidence. Further studies of direct measurement of histone deacetylation status (i.e. acetylated histone levels) and HDAC enzymatic activity will be needed to strengthen the mechanistic interpretation.

  1. (MAJOR) While MAGT1 was upregulated, direct evidence of its role in Mg2+ transport and subsequent CCL2 regulation (MAGT1 knockdown) is not studied. Include experiments that directly test the involvement of MAGT1, such as siRNA knockdown of MAGT1, to confirm its role in Mg2+-induced CCL2 upregulation.

According to the reviewer’s comments, we examined the effect of siRNA-mediated inhibition of MAGT1 on high [Mg2+]e-induced upregulation of CCL2 in M2-MACs. As expected, MAGT1 inhibition reduced the CCL2 level in M2-MACs. The obtained results were shown in Figures 9E/9F and the ‘Results’ section. (Page 6, line 262-268; Page 11, line 371-Page 12, line 373)

  1. In figure 1, the y-axis labels "ratio to ACTB" (A, B) and "relative CCL22 mRNA level" / "relative CCL2 mRNA level" (C, D, F, G) are inconsistent. It would be clearer to use a unified "Relative mRNA expression (normalized to ACTB)" or "Fold change vs. control" for all.

As described above (Responses to Reviewer 3, Results, Comment #2), we intentionally did not use the same y-axis scale because we would like to express the target gene expression as a ratio to actin when using it for the first time. We added the fold change between native and M2-MACs in the text (Section 2.1). (Page 3, line 93-94)

  1. In figure 2, the notation in the x-axis "KCl +/‒/‒", "+/+/‒" etc. is somewhat difficult to understand. A legend defining these combinations ("‒": vehicle, "+": 30mM KCl, "SKA": SKA121, "EDV": EDV) within the figure caption or panel is necessary. Also, consider showing absolute intracellular K+ concentrations, or at least a standard curve, to provide more quantitative context.

11-1. We agree with the reviewer’s indication. We added the following description in the legends of Figure 2.

The plus (+) and minus signs (−) are used to represent the presence and absence of the reagents, respectively.

11-2. From a standard curve, we calculated the intracellular K+ concentration per mg protein: 2.673 +/- 0.053 mM/mg protein in control (‒/‒) (n = 4) (Fig. 2D), which is estimated between 140 and 150 mM. We added this in the ‘Results’ section (Section 2.2) (Page 3, line 137-Page 4, line 139). In addition, we added the values of CCL2 concentration in control (//‒) and 35 mM [K+]e groups (+//) (Sention 2.2) (Page3, line 121-123).

  1. In figure 5 and figure 6, label the lanes in D sections for K+ concentration. Also, it seems like that the ACTB expression as a HKG is not consistent in all experiments. Clarify.

We conformed that labels in Figs. 5D and 6D; however, the labels for K+ concentration were correct. We will confirm the uploaded Word file after resubmission.

As described in Section 4.4, the protein expression levels of the target genes (CEBPB and CREB2 in Figs. 5 and 6) were compensated by the expression levels of ACTB (β-actin), a house keeping gene, and were then quantitated.

Discussion

  1. The discussion acknowledges the limitation of using THP-1 cells, but it doesn't sufficiently discuss how this model limits the findings or what specific differences might exist between THP-1 derived M2-MACs and in vivo TAMs. For example, TAMs are highly heterogeneous and influenced by various factors in the TME not replicated in this model. Enrich this section.

We simply described the limitation in THP-1-differentiated M2 macrophages as a model of TAMs in the ‘Discussion’ section (Section 3) (Page 14, line 453-455 in the original version). According to the reviewer’s comment, we further added the description. (Page 15, line 496-Page 16, line 499)

In vitro–differentiated M2 macrophages do not fully recapitulate the diverse intracellular signaling of TAMs and lack exposure to in vivo TME conditions such as hypoxia, elevated lactate, and low glucose.

  1. Several connections, such as the potential involvement of RUNX1-HDAC3/CREB2/CEBPB axes or specific miRNAs, are introduced based on existing literature but not directly investigated. While good for future research, they should be presented as hypotheses more clearly.

As for the involvement of RUNX1 in CCL2 transcription, we performed the additional experiments using siRNA for RUNX1. We added the data in Supplementary Fig. S5, amended the description in the ‘Discussion’ section (Section 3), and presented as hypotheses. (Page 15, line 468-474)

Additionally, according to the reviewer’s suggestion, the possible involvement of specific miRNAs is presented as a hypothesis. (Page 15, line 461-462)

  1. Propose concrete next steps using primary human TAMs, co-culture systems with tumor cells, or relevant in vivo animal models to validate the findings. Discuss the complexities of TAM heterogeneity and how this in vitro model might misrepresent it.

According to the reviewer’s suggestion, we added the following sentences in the ‘Conclusions’ section (second half of Section 5).

The current findings, derived from in vitro experiments using THP-1-differentiated M2 macrophages, provide valuable insights into the regulatory mechanisms involved in CCL2 expression under tumor-mimicking ionic conditions. As a next stage, it will be important to assess whether similar regulatory pathways operate in primary human TAMs, co-culture systems involving tumor cells and macrophages, and in vivo tumor models that recapitulate the complexity of the TME. These validation studies will help to confirm the translational relevance of our findings and support their potential application in therapeutic strategies targeting TAM-mediated immunosuppression.

Materials and Methods

  1. The timing of drug applications (48- or 60-hr after IL-4/IL-13 incubation) should be more justified in relation to the differentiation state of the M2-MACs. Are they fully differentiated and stable by this point?

Several research groups have conducted experiments differentiating THP-1 cells into M2-like macrophages. They stimulated THP-1 cells with PMA for 8–12 hr, followed by IL-4 and IL-13 stimulation for an additional 24- to 72-hr, and demonstrated that M2-like macrophage differentiation occurs as early as 24-hr after IL-4/IL-13 treatment. In our study, we applied the same stimulation protocol and confirmed that M2 markers such as CD163, arginase 1, and IL-10 were highly expressed 24-hr after IL-4/IL-13 stimulation and remained stably elevated up to 72-hr. Therefore, to align the endpoints for drug application, we added drugs 12-hr before RNA collection (at 60-hr) and 24-hr before protein collection (at 48-hr). We added the description of this in the ‘Materials and Methods’ section (Section 4). (Page 17, line 563-567) Thank you for your careful reading and adequate advice.

  1. (MAJOR) For the confocal imaging, clarify whether 'n' refers to biological replicates. If not, multiple independent experiments should be performed, with results from each replicate quantified and analyzed.

As indicated by the reviewers, we should clarify the biological replicates for the confocal imaging. According to the other reviewers’ comments, for real-time PCR, Western blots, and ELISA, we added the descriptions on biological replicates to the ‘Materials and Methods’ section (Section 4).

For confocal imaging, we noticed the errors in the description of the data acquisition and amended it as follows,

For each experimental replicate (n =1), fluorescence intensities were obtained from at least four frame images, containing more than 30 cells. Summarized data were derived from six independently differentiated M2-MAC samples. Cell immunostaining, imaging acquisition, and quantitative analyses were independently performed by separate investigators to minimize potential bias.

We added these sentences in Section 4.7. (Page 18, line 609-613)

  1. Include full supplier names and locations for all reagents and kits for maximum reproducibility. Also, specify the software versions used for densitometry and real-time PCR analysis.

According to the reviewer’s indication, we described the full supplier names and locations for all reagents and kits, and specified the software versions used for densitometry and real-time PCR analysis (see Section 4).

 The English could be improved to more clearly express the research.

According to the reviewer’s instructions, the English was carefully revised to make the research findings clearer by all authors.

Round 2

Reviewer 2 Report

Comments and Suggestions for Authors

The comments/concerned are addressed properly.

English: the revised abstract is found to contain a minor glitch (Kca3.1 and LRRC8A activities), so there may be more sentences to be improved.

Author Response

Thank you very much for your thoughtful and careful review. In response to your comments on the Abstract, we have made the necessary revisions and also re-examined the entire manuscript.

Reviewer 3 Report

Comments and Suggestions for Authors

The authors have comprehensively addressed all my comments, leading to substantial improvements in the manuscript. Specifically, limitations concerning siRNA efficiency and the use of THP-1 cells are acknowledged and discussed, with suggestions for future gene-editing approaches and primary cell models. The independent action of LRRC8A inhibition from the WNK1-AMPK-Nrf2 axis is emphasized. A crucial point on directly assessing HDAC3 activity is addressed, with future experimental directions outlined. Experiments involving MAGT1 siRNA knockdown have been performed and added. Finally, the English language has been carefully revised for clarity. Accordingly, the article is now suitable for publication.

Author Response

Thank you very much for your thoughtful and careful review.